# PRAME-AS lncRNA, regulated by MZF1, modulates PRAME expression and cell stemness

**Zahra Hosseininia**[1,2]**, Hesam Dehghani**[1,2,3]*

**1** Division of Biotechnology, Faculty of Veterinary Medicine, Ferdowsi University of Mashhad, Mashhad, Iran, **2** Stem Cells and Regenerative Medicine Research Group, Research Institute of Biotechnology, Ferdowsi University of Mashhad, Mashhad, Iran, **3** Department of Basic Sciences, Faculty of Veterinary Medicine, Ferdowsi University of Mashhad, Mashhad, Iran

* dehghani@um.ac.ir

## Abstract

PRAME is a member of the large cancer-testis antigens (CTA). In normal tissues, PRAME is mainly expressed in the testis. However, low levels of PRAME transcript are detected in the endometrium, ovary, and adrenal glands. Due to the tumor-restricted expression of PRAME, some of its regulatory factors have been investigated; however, the role of lncRNAs remains unclear. In this study, we ablated the *PRAME-AS* gene and investigated the effects on PRAME expression, cell proliferation, migration, viability, and anchorage-independent growth. We further manipulated PRAME-AS expression through MZF1 overexpression and hypermethylation of its upstream regulatory region. Additionally, we assessed the ability of the PRAME locus regulatory region to drive PRAME-AS lncRNA transcription. Our findings show that *PRAME-AS* gene knockout diminishes PRAME transcript levels by about 37%. The *PRAME-AS* knockout cells showed a decrease in migration, proliferation, stemness, and viability. We discovered that MZF1 induces PRAME expression approximately twofold and increases PRAME-AS transcript levels by more than threefold. We found that hypermethylation of the upstream regulatory sequence of PRAME-AS results in an approximately 50% reduction in PRAME transcripts and an 18.5% decrease in PRAME-AS transcript levels. Moreover, based on our EGFP reporter assay, we conclude that the regulatory region of the PRAME locus—without PRAME transcription—is insufficient to drive transcription of PRAME-AS in the antisense direction. Taken together, our data support the conclusion that PRAME-AS lncRNA acts as a regulator of PRAME transcript levels, and that MZF1—and the methylation status of a shared CpG island— influence the expression of both genes.

## Introduction

PRAME (PReferentially expressed Antigen in Melanoma) is also known as MAPE (Melanoma Antigen Preferentially Expressed in tumors), OIP4 (OPA-Interacting

provided the original author and source are credited.

**Data availability statement:** All relevant data are within the manuscript and its Supporting information files.

**Funding:** This work was supported by a Ferdowsi University of Mashhad (http://en.um. ac.ir/) grant to HD (No. 53395). The funders had no role in study design, data collection and analysis, decision to publish, or preparation of the manuscript. There was no additional external funding received for this study.

**Competing interests:** The authors have declared that no competing interests exist.

Protein 4), and CT130 [1]. PRAME, along with three protein-coding genes, *GGTLC2* (gamma glutamyltransferase light chain 2), *ZNF280A* (zinc finger protein 280A), and *ZNF280B* (zinc finger protein 280B), and a microRNA gene (*miR-650*) are located in a segment of immunoglobulin lambda light chain (IGL) on chromosome 22 at locus 22q11.22 [2]. It is transcribed from an approximately 12-kilo base pair genomic region and produces 14 transcript variants (Ensembl: ENSG00000185686) [3]. PRAME is a member of the large cancer-testis antigen (CTA) family. CTA members are expressed in human malignancies with different histological origins, although they are not detected in normal tissues except for the testis and placenta [4]. Because of tumor-restricted expression of CTAs, they might be responsible for modulating tumor gene expression and cell division [5].

Similar to other CTAs, PRAME is expressed in the normal testis. However, low levels of PRAME transcript are detected in the endometrium, ovary, and adrenal glands [6]. PRAME was first reported as a tumor antigen that induces immune responses by stimulating cytotoxic T lymphocytes in metastatic cutaneous melanoma [7]. In the subsequent reports, its increased expression at both mRNA and protein levels (S1 Fig) was confirmed in different kinds of cancers, including breast cancer [8], hepatocellular carcinoma [9], osteosarcoma [10], primary and metastatic uveal melanoma [11–13], hematological malignancies [14,15], lung cancer [16–18], prostate cancer [19], seminomas [20], and cervical cancer [21].

PRAME has been implicated in promoting different malignancies by affecting fundamental cellular mechanisms, such as cell cycle, differentiation, apoptosis, EMT, and pluripotency. Thus, regulating its expression has been the subject of some research reports. For example, Nettersheim et al. reported the PRAME expression in seminomas following the upregulation of SOX17, along with the enrichment of H3ac at its transcription start site (TSS), and decreased methylation of its promoter [20]. Hypomethylation of specific CpG sites at the 5′ regulatory region of *PRAME* corresponds to its high expression in different tumor cell lines, tissues, and clinical AML samples [22]. Hypomethylation and myeloid zinc finger 1 (MZF1) overexpression promote PRAME expression in A375P melanoma cells [23]. Also, in a melanoma cell line, it has been shown that the down-regulation of microRNA-211 results in PRAME up-regulation by affecting 2 binding sites in the PRAME 3′ untranslated region (3′-UTR) [24]. Moreover, melanoma cells with SOX9 overexpression restore sensitivity to RA treatment as a result of decreasing PRAME expression [25]. Ectopic expression of miR-421 in human prostate cancer cell lines by direct interaction with PRAME mRNA leads to its suppression at both transcript and protein levels [19]. In recent studies, it has become apparent that PRAME is up-regulated following Gas6 stimulation, which is mediated through Axl/ERK1/2 signaling pathway in hepatocellular cancers [26]. Besides, loss of HDAC5 induces PRAME expression in laryngeal squamous cell carcinoma (LSCC) [27].

In this study for the first time, we questioned the role of PRAME-AS (PRAME antisense, LL22NC03-63E9.3) lncRNA in PRAME expression. We knocked out the *PRAME-AS* gene using our previously reported TransCRISTI method [28]. We also manipulated PRAME-AS expression using two approaches: its upregulation

via MZF1 overexpression and its downregulation by epigenetic modification of the upstream regulatory region using dCas9-DNMT3A. The *PRAME-AS* knock-out cells showed decreased PRAME transcript levels and diminished cell viability, proliferation, migration, and stemness properties. Our findings show that the MZF1 transcription factor promotes both PRAME and PRAME-AS expression levels. Consistent with these observations, hypermethylation of the *PRAME-AS* upstream regulatory region negatively affects transcript levels of both genes. Overall, our data suggest that PRAME-AS lncRNA can be regarded as a regulator of PRAME transcript levels.

## Materials and methods

### Bioinformatic analysis of PRAME and PRAME-AS lncRNA transcripts

To analyze the expression profile of PRAME and PRAME-AS transcripts in cancer and normal tissues, we used the Gene Expression Profiling Interactive Analysis (GEPIA2) web-based tool (http://gepia.cancer-pku.cn/), which uses The Cancer Genome Atlas (TCGA) and the Genotype-Tissue Expression (GTEx) data [29]. We used the expression DIY function to compare PRAME and PRAME-AS lncRNA transcript levels between normal and cancer states. We also used the LinkedOmicsKB web tool (https://kb.linkedomics.org/) to derive PRAME protein expression levels in tumor and normal tissues.

### Plasmid constructs

To perform *PRAME-AS* gene knockout and to assess the effects of DNA methyltransferase 3A (DNMT3A) on the CpG island in the regulatory region of the *PRAME-AS* gene, appropriate single guide RNA (sgRNA) sequences were designed (Table 1) using the CHOPCHOP online software (https://chopchop.cbu.uib.no). To design sgRNAs, we avoided the overlapping sequences of the 5′ region of the *PRAME-AS* gene and the 5′ region of the *PRAME* gene. We also tried to design sgRNAs that may not recognize genomic off-targets.

In order to induce methylation of a CpG island in the regulatory region of the PRAME-*AS* gene, we mapped the exact position of predicted CpG dinucleotides according to Schenk et al. [22]. In the next step, two sgRNAs were designed to target dCas9-DNMT3A molecules to this island. According to Vojta et al. [30], a 25–35 bp region downstream of the proto-spacer adjacent motif (PAM) can be an optimum range for an effective methylation induced by dCas9-DNMT3A and, the CpG dinucleotides outside the optimum range of 25–35 bp from the PAM sequence are less likely to become methylated. sgRNA coding sequences were synthesized by annealing oligonucleotides (Macrogen Inc., Seoul, South Korea) in a reaction containing one µl of each oligonucleotide (100µM), and 1X T4 DNA Ligase buffer (Thermo Fisher Scientific, Waltham, MA, USA), and incubating in a thermocycler at 95°C for 3 minutes followed by downward gradient toward 25 °C (1 minute at each thermal point and 5 minutes at 25 °C). The annealing of two oligonucleotides produced a double-stranded DNA fragment with two BbsI sticky ends matching those of the BbsI-digested pHD_5009 [28] and Addgene #71666 [30] plasmids.

To generate the pHD_4091 vector (Table 2), the coding sequence for the MZF1 transcription factor, along with a part of the CMV promoter and FLAG sequence, was cut by NdeI and XbaI from the FLAG-tagged MZF1 pcDNA3 (a gift from Prof. Soo-Jong Um, Department of Integrative Bioscience and Biotechnology, Sejong University) and sub-cloned into the NdeI/NheI sites of the pHD_4090 vector containing 5′TR-CMV-MCS-PA-EF1a- puroR-T2A-IRES-EGFP-3′TR. The sequence and open reading frame (ORF) of the MZF1 were confirmed by Sanger sequencing (Microsynth Inc., Balgach, Switzerland) (S2 Fig).

To further study a predicted region containing the upstream sequences of PRAME [22], we amplified a 1361 bp sequence from the genome of wild-type Human Embryonic Kidney (HEK) 293 T cells and cloned it in two opposite directions into the EcoRI/SalI and SacI/SalI sites of the pHD_4016 vector replacing the CAG promoter. With this sequence (S1 File), named here the "PRAME locus regulatory region" two constructs were generated; pHD_4016-1, and pHD_4016-2 (verified by Sanger sequencing; Microsynth Inc., Balgach, Switzerland) (S3 and S4 Figs). In addition, by removing the

**Table 1. Oligonucleotides used in this study.**

| Primer name | Primer sequence (5′_3′) | Product length (bp) | Application |
|---|---|---|---|
| F1 | CACCGTCCATTCTCAACCCTTAGG | – | Making sgRNA-encoding sequences to target (Knock-out) the *PRAME-AS* gene (NCBI Gene ID: 648691) |
| R1 | AAACCCTAAGGGTTGAGAATGGAC | – | |
| F2 | CGGGCCATTTACCGTAAG | 509 | PCR of genomic DNA in the PRAME-AS locus (NCBI Gene ID: 648691) |
| R2 | TGCTATGGTTGTCCCAGAAC | 509 | |
| F3 | GCTCGGGACTTACATCGGTC | 212 | Quantification of PRAME transcripts (NCBI Accession: NM_006115.5) |
| R3 | CCTGCAGGCTCTCTATGTGG | 212 | |
| F4 | TAGGGGGATGGTCAGGCTTC | 312 | Quantification of PRAME-AS transcripts (NCBI Accession: NR_027426.2) |
| R4 | ATGCACTAAGTGTCGTGGGA | 312 | |
| F5 | CCTCGCCTTTGCCGATCCG | 257 | Quantification of β-actin transcripts (NCBI Accession: NM_001101) |
| R5 | GGTACTTCAGGGTGAGGATGC | 257 | |
| F6 | CACCGGCTGAAGAGACCACCCCCC | – | Making sgRNA-encoding sequences (sgRNA1 for PRAME-AS methylation) |
| R6 | AAACGGGGGGTGGTCTCTTCAGCC | – | |
| F7 | AAACCTCTCCACAGAAATCCACGC | – | Making sgRNA-encoding sequences (sgRNA2 for PRAME-AS methylation) |
| R7 | CACCGCGTGGATTTCTGTGGAGAG | – | |
| F8 | GTGATTTGTTAATAGGTTTGTATTGG | 368 | Bisulfite sequencing |
| R8 | CAAAATCCCCTAAACTCAAATTCTC | 368 | |

CAG promoter between SalI/XhoI sites of the pHD_4016 plasmid, a promoterless pHD_4016-3 construct was created (Table 2).

## Cell culture and transfection

HEK293T cell line was purchased from the Pasteur Institute Cell Bank (Tehran, Iran). This cell line was cultured in high-glucose Dulbecco's Modified Eagle Medium (DMEM; Gibco, Grand Island, NE, USA) supplemented with 10% heat-inactivated fetal bovine serum (FBS; Gibco, Grand Island, NE, USA) and 1% penicillin-streptomycin (Gibco, Grand Island, NE, USA). Twenty-four hours before transfection, around 70,000 cells were seeded in each well of the 24-well plates and cultured at 37 °C in a 5% $CO_2$ atmosphere. Cell transfections were carried out in three biological replicates in all experiments. The polyplex (the complex of bPEI25 polymer-plasmid DNA) was mixed with no serum DMEM medium to a total volume of 500 µl. After 10 minutes of incubation at room temperature, this mixed solution was added to the cells. After 150 minutes, the transfection medium was replaced with the medium with 10% heat-inactivated FBS without penicillin-streptomycin. The details of transfections for various applications, including gene knockout, gene overexpression, regulatory region methylation, and PRAME locus regulatory region assay, are provided in S1 Table.

## TransCRISTI method

To ablate the *PRAME-AS* lncRNA gene, we used our previously developed TransCRISTI method [28]. Briefly, HEK293T cells were transfected (using 25 kDa branched Polyethylenimine; bPEI25, Sigma-Aldrich, USA) with Cas9.PB^dm (Cas9 fused to double mutant *piggy*Bac transposase) and sgRNA #1 encoding plasmid (pHD_5009-1), and a donor plasmid (pHD_4012; 5′ TR-CAG promoter-PuroR-IRES-EGFP-PA-3′TR) (S1 Table). After 2 hours of incubation with polyplex, the transfection medium was replaced with the medium containing 10% FBS and no antibiotics. Forty-eight hours post-transfection, cells were selected for 3 days in the medium containing 0.7 µg/ml puromycin sulfate (Sigma-Aldrich, USA), followed by a 3-day recovery (medium without puromycin sulfate). Periods of puromycin treatment and recovery were continued until we had an enriched EGFP-expressing population of cells.

**Table 2. Plasmid constructs used in this study.**

| Construct name | Features | Application | Reference |
|---|---|---|---|
| pHD_5009–1 | CAG-Cas9.PBdm-PA | Encoding dCas9.PB & sgRNA1; used to Knockout the target gene | This study |
| pHD_4012 | 5′TR-CAG-puroR-IRES-EGFP-PA-3′TR | Donor plasmid; used to knockout the target gene | [31] |
| pHD_4090 | 5′TR-CAG-MCS-PA- EF1α- PuroR-T2A-IRES-EGFP-3′TR | Encoding EGFP & puromycin N-acetyltransferase; used as negative control for MZF1 overexpression | This study |
| pHD_4091 | 5′TR-CAG-FLAG.MZFI-MCS-PA- EF1α-PuroR-T2A-IRES-EGFP-3′TR | Encoding MZF1, EGFP, & puromycin N-acetyltransferase; used to induce MZF1 overexpression | This study |
| pHD_3501 | CMV-mPB.TP-bGH-PA | Encoding *piggyBac* transposase; used to induce transposition | [31] |
| Addgene #71666 | U6-CMV-3Flag-NLS-dCas9-NLS-DNMT3a-T2A-EGFP-PA | Encoding dCas9-DNMT3a & EGFP; used as negative control for hypermethylation | [30] |
| pHD_71666–1 | U6-sgRNA1-CMV-3Flag-NLS-dCas9-NLS-DNMT3a-T2A-EGFP-PA | Encoding dCas9-DNMT3a, EGFP, & sgRNA1; used to induce hypermethylation | This study |
| pHD_71666–2 | U6-sgRNA2-CMV-3Flag-NLS-dCas9-NLS-DNMT3a-T2A-EGFP-PA | Encoding dCas9-DNMT3a, EGFP, & sgRNA2; used to induce hypermethylation | This study |
| pHD_4016–1 | PRAME locus regulatory region (PRAME Dir.) -PuroR-IRES-EGFP-PA | Expressing EGFP under the control of the PRAME locus regulatory region in the PRAME direction | This study |
| pHD_4016–2 | PRAME locus regulatory region (PRAME-AS Dir.)-PuroR-IRES-EGFP-PA | Expressing EGFP under the control of the PRAME locus regulatory region in the PRAME-AS direction | This study |
| pHD_4016–3 | PuroR-IRES-EGFP-PA | Promoterless plasmid, used as a negative control | This study |

## DNA extraction and genomic DNA PCR

Genomic DNA was isolated from cells using the Animal Tissue DNA Isolation kit (DENAzist Asia Co., Mashhad, Iran). PCR reactions on genomic DNA consisted of 1µl of each 10µM primer (Table 1), 2 µl DNA, 10 µl Taq DNA Polymerase 2x Master Mix RED (Ampliqon, Odense, Denmark) in a final volume of 20 µl. PCR reactions were performed in a Bio-Rad T100 Thermal Cycler (Bio-Rad, Hercules, California, USA) and contained an initial denaturation at 95°C for 3 min, followed by 35 cycles of denaturation at 95°C for 30 s, annealing at 55 °C for 30 s, elongation at 72°C for 20 s, and a final extension of 72 °C for 10 min. Ten µL of the PCR product was analyzed on a 1% agarose gel by electrophoresis. The PCR amplicon was verified by SacI restriction enzyme digestion (Thermo Fisher Scientific, Waltham, MA, USA) and Sanger sequencing (Microsynth Inc., Balgach, Switzerland).

## Bisulfite conversion assay

To design primers for the methylated region, we used the Bisulfite Primer Seeker online tool (https://www.zymoresearch.com/pages/bisulfite-primer-seeker) to convert the target region to a bisulfite-treated sequence. The non-CpG cytosines in the target DNA sequence were converted to uracil, while CpG cytosines were kept unchanged. In the next step, the bisulfite-converted sequence was subjected to primer design using EpiDesigner online software (http://www.epidesigner.com). We considered specific criteria, including choosing the correct template strand, avoiding CpG islands on the sequence of primers, higher annealing temperatures, longer primer lengths, and an amplicon size between 150–400 bp. Then, the specificity of primers was checked by the Bisearch online tool (http://bisearch.enzim.hu/?m=search) to assess the non-specific annealing of primers in the bisulfite-treated genome, and the NCBI genomic BLAST tool (https://www.ncbi.nlm.nih.gov/tools/primer-blast) to assess the probable non-specific binding throughout the genome. The primers were synthesized by Generay Inc., (Shanghai, China).

Seventy-two hours after transfection, cells were subjected to DNA extraction using the Animal Tissue DNA Isolation Kit (DENAzist Asia, Mashhad, Iran). The isolated genomic DNA was bisulfite converted using Thermo Scientific™ EpiJET™

Bisulfite Conversion Kit (Waltham, Massachusetts, USA). The PCR reaction contained 0.2 µl of bisulfite-treated DNA, 10 µl of Taq DNA Polymerase 2x Master Mix RED (Ampliqon, Odense, Denmark), and 1µl of each of 10 µM specific primers (Table 1) in a 20 µl reaction. The thermocycling program included a cycle of 95 °C for 3 min; 35 cycles of 95 °C for 30 s, 60 °C for 30 s, and 72 °C for 20 s, followed by a final extension of 72 °C for 10 min in a Bio-Rad T100 Thermal Cycler (Bio-Rad, Hercules, California, USA). A 368 bp PCR product was subjected to Sanger sequencing (Microsynth Inc., Balgach, Switzerland).

### RNA extraction, reverse transcription, and quantitative polymerase chain reaction (qPCR)

Total RNA was extracted from different cell lines using the Column RNA Isolation kit (DENAzist Asia, Mashhad, Iran). The quality and quantity of RNA were analyzed using gel electrophoresis and a NanoDrop Epoch2 microplate spectrophotometer (BioTek, Winooski, Vermont, USA). The DNase treatment of 5 µg total RNA was performed in the presence of 4 µl DNase enzyme (Sinaclon, Tehran, Iran) in a total volume of 20 µl. The reaction was incubated at 37 °C for 25 min. Then, the enzyme was inactivated by adding 2.5 µl 0.2 M EDTA and heating at 65 °C for 10 min. First-strand cDNA reaction contained 4–5 µg of total RNA, 1 µl of 100 µM oligo(dT)18 Primer (DENAzist Asia, Mashhad, Iran), 1X RT buffer, 2 µl of dNTPs (10mM each, DENAzist Asia, Mashhad, Iran), and 200 U of reverse transcriptase (RT; Thermo Scientific, Waltham, Massachusetts, USA) in a total volume of 20 µL. The reaction was incubated for 60 min at 42 °C in a Bio-Rad T100 Thermal Cycler (Bio-Rad, Hercules, California, USA).

The PCR reaction for amplification of PRAME-AS was carried out using 3µl of cDNA, 1µl of F2 and R2 primers (Table 1), and 10 µl of Taq DNA Polymerase 2x Master Mix RED (Ampliqon, Odense, Denmark). The PCR program comprised one step of 95 °C for 3 min, followed by 40 cycles of 95 °C for 30 s, 55 °C for 30 s, and 72 °C for 30 s, and a final extension at 72 °C for 10 min.

qPCR reactions for quantification of PRAME and PRAME-AS transcripts contained 2 µl cDNA, and forward and reverse primers (Table 1) in the presence of the RealQ Plus 2x Master Mix Green (Ampliqon, Odense, Denmark) in a Rotor-Gene Q Real-Time PCR Analyzer (Serial No: 0512179, Qiagen, Hilden, Germany). Reactions were performed in three technical replicates. The qPCR program for PRAME and ACTB started with 15 min initial denaturation at 95 °C, and continued by 40 cycles of 30 s melting at 95 °C, 30 s annealing at 60 °C, and 20 s extension at 72 °C. The qPCR program for PRAME-AS transcript started with an initial denaturation at 95 °C for 15 min, followed by 45 cycles of 95 °C (30 s), 68 °C (30 s), and 72 °C (8 s). Each cycle ended with 10 s of acquiring green fluorescence at 75 °C for PRAME and ACTB and at 80 °C for PRAME-AS. qPCR programs were finished by generating melting curves, ramping from 50 °C to 99 °C, 5s acquiring green fluorescence at each °C (S5 Fig). Also, the identity of amplification products was confirmed by gel electrophoresis (S5 Fig). The Pfaffl method [32] was used to quantify relative differences between groups. qPCR analyses were carried out in three biological replicates.

### Immunocytochemistry and DAPI staining

To confirm the overexpression of the MZF1 protein, we used immunocytochemistry (ICC) to detect FLAG-MZF1 fusion protein. The FLAG-MZF1 overexpressing and control cells were cultured on 1% gelatin-coated glass coverslips in a 6-well tissue culture plate. The cells were incubated at 37 °C in a humidified 5% $CO_2$ incubator until they reached 50–60% confluency. At first, the culture medium from each well was aspirated, and gently, the cells were washed with warm (37 °C) 1X D-PBS for 5 min at room temperature (RT). Then, cells on coverslips were fixed with 4% paraformaldehyde in 1X D-PBS for 10 min and rinsed three times with 1X D-PBS (5 min each time). Next, the cells were permeabilized by 0.5% Triton X-100 (Merck, Darmstadt, Germany) in 1X D-PBS for 5 min at RT and washed with 1X D-PBS (three times, 5 min each). After that, the non-specific proteins were blocked by exposing the cells to 3% bovine serum albumin (BSA) for 1 hour at RT. After three washes with 1X D-PBS (5 min each), cells were

incubated overnight with the anti-FLAG tag primary antibody (Cat. No. 100233-MM01, 1:200, Sino Biological Inc., Beijing, China) in 1X D-PBS at 4°C. The next day, cells were washed three times with 1X D-PBS (5 min each) and incubated for 2 hours at RT in anti-mouse IgG, HRP-linked secondary antibody (Cat. No. 7076P2, 1:500, Cell signaling technology, Danvers, Massachusetts, USA) in 1% BSA. Cells were washed with 1X D-PBS three times, 5 min each, and stained with 1% 3,3′-diaminobenzidine (DAB, Cat. No. D8001-1G; Sigma-Aldrich, St. Louis, Missouri, USA) at RT for 5 min. Following a wash with 1X D-PBS, the coverslips were placed (cell side down) on a slide with a drop (50–100 µl) of antifade + DAPI (1 µg/ml, Cat. No. S-8016, DENAzist Asia Co., Mashhad Iran) to stain the nuclei. Finally, the coverslips were sealed with nail polish and were analyzed under a Nikon Eclipse Ts2 inverted microscope (Nikon Co., Tokyo, Japan).

## MTT assay

A total of $7 \times 10^3$ cells were seeded per well of a 96-well tissue culture plate in 9 technical replicates. After 48h, MTT (3-(4,5-Dimethylthiazol-2-yl)-2,5-Diphenyltetrazolium Bromide) (Sigma-Aldrich, St. Louis, Missouri, USA) was added to a final concentration of 0.5 mg/ml into the culture medium. Cells were incubated at 37 °C for 3 hours in the dark. At the end of this period, the formazan precipitates were observed under a microscope. Then, the media was removed. To dissolve formazan crystals, 100 µl DMSO was added to each well and incubated at room temperature in the dark for 15 min. Finally, the purple color was measured at 570 nm using an Epoch 2 microplate reader (BioTek, Winooski, Vermont, USA). This experiment was performed in three biological replicates.

## Colony formation assay

Two thousand four hundred cells were seeded per well of each 6-well tissue culture plate in 2 technical replicates. The cells were cultured at 37 °C in a humidified 5% CO2 incubator for 9 days. Fresh culture medium was added to the wells every 3 days during this period. On the ninth day, when cells formed sufficiently large clones, the medium above the cells was removed. After washing the cells with warm 1X D-PBS (37°C), a mixture of 0.5 ml of 4% paraformaldehyde in 1X D-PBS and 0.5 ml of 0.5% methylene blue in 50% ethanol was added to each well and kept at RT for 30 min [33]. Then, cells were washed with 1X D-PBS three times (5 min each). The experiment was carried out in two biological replicates. In the end, the colonies in each well were counted and photographed using an inverted microscope. The diameter of colonies was measured using ImageJ software [34].

## Soft agar assay

To analyze the ability of anchorage-independent growth in cells, 12 ml of DMEM/F12 supplemented with 5% horse serum and 1% penicillin/streptomycin was added to 3 ml of 0.3% 2-hydroxyethyl agarose (Sigma-Aldrich, St. Louis, Missouri, USA). Then, 2 ml (per well) of this mixture was poured into each well of a 6-well tissue culture plate as the bottom layer. The plate was kept in the laminar flow hood to cool for 15 min. Subsequently, the plate was incubated in a refrigerator for 1 hour. The top (cell) layer was prepared by a high agar concentration (0.6% 2-hydroxyethyl agarose, Sigma-Aldrich, St. Louis, Missouri, USA) and was appropriately mixed with $2 \times 10^5$ cells. The cell layer was added gently to prevent disruption of the bottom layer. After pouring the cell layer, the plate was placed into the refrigerator for about 10 min to accelerate the solidification. The cells were cultured at 37°C in a humidified 5% $CO_2$ incubator for one week. The fixation and staining procedure were performed when cells formed sufficiently large clones. A mixture of 0.5 ml of 4% paraformaldehyde in 1X D-PBS and 0.5 ml of 0.5% methylene blue in 50% ethanol was added to each well and kept at RT for 30 min. Then, cells were washed with 1X D-PBS thrice (5 min each) [33]. The experiment was carried out in two biological replicates. In the end, the colonies were counted and photographed under an inverted microscope. The diameter of colonies was measured using ImageJ software [34].

## Wound healing assay

A lower percentage of serum in the growth media during the wound healing assay has been suggested to differentiate proliferation from migration. However, the amount of serum for every cell type must be adjusted so that its reduction does not promote apoptosis or cell detachment [35]. Therefore, $6 \times 10^4$ wild-type HEK293T cells were plated in each well of a 24-well tissue culture plate and maintained at 37 °C in a humidified 5% $CO_2$ incubator for 72h. When the cells were uniformly spread and gaps between cells were minimized, the normal growth medium was exchanged with DMEM medium containing various FBS percentages (0%, 1%, 2%, 3%, 4%, and 10%) (S6 Fig). We found that 2% FBS is the most appropriate concentration for performing the wound healing assay on the HEK293T cell line. After incubation for 24 hours, the monolayer of cells was scratched with a 10 μl pipette tip. The medium was immediately changed to remove cell debris and mitogenic stimuli released from the ruptured cells [36]. The scratch area was marked on the bottom of the plate. The open gaps (the scratch area) were imaged at 0h (the time of creating the scratch) and 24h (24h after creation of the scratch) under an inverted microscope and analyzed by ImageJ software. The experiments were performed two times, each with three technical replicates. The percentage of wound closure was calculated according to the following equation [37]:

$$Wound\ closure\ \% = \frac{(A_0 - A_{24})}{A_0} \times 100$$

where $A_0$ and $A_{24}$ represent the scratch area at 0h and 24h after creating the scratch, respectively.

## Flow cytometry

Forty-eight hours post-transfection, transfected cells were dissociated using trypsin/EDTA solution (Biowest, Nuaillé, France). Detached cells were washed twice with and were resuspended in 0.5 ml of cold D-PBS. The samples were analyzed to determine the percentage of EGFP-positive cells by FACS Calibur (BD Biosciences, Franklin Lakes, New Jersey, USA). The data were analyzed by the FlowJo version 10 software (FlowJo software, Ashland, Oregon, USA). The experiment was executed in three biological replicates, each in two technical replicates.

## Statistical analysis

In this study, all data were summarized and analyzed using GraphPad Prism 8.0 (GraphPad Software, Inc., San Diego, CA, USA). The size of colonies in soft agar and colony formation assays, scratch areas, and the mean intensity of green fluorescence were measured using ImageJ software. The statistical differences for nonparametric data including, MTT, wound healing assays, qPCR data, the percentage of fluorescent cells, the mean intensity of green fluorescence, and flow cytometry findings, were analyzed by the Mann-Whitney U-test. The Welch correction test was used for parametric data (soft agar and colony formation assays).

## Results

### PRAME and PRAME-AS lncRNA transcript levels differ between cancerous and normal states of solid tissues

Investigation of the TCGA and GTEx datasets by the GEPIA2 web server showed that in solid tumors, the transcript number of PRAME is significantly higher than that of the corresponding normal tissues (Fig 1B and S7 Fig). While the number of PRAME-AS lncRNA transcripts was generally higher in some solid cancers (skin, lung, ovary, and uterus) than in their corresponding normal tissues, the differences were insignificant (Fig 1C and S7 Fig). Normal testicular tissue had high levels of PRAME and PRAME-AS lncRNA transcripts (Fig 1 and S7 Fig).

### Ablation of the *PRAME-AS* gene affects PRAME transcript levels

To determine the possible effect of PRAME-AS lncRNA on PRAME expression (Fig 2A), we decided to knockout the *PRAME-AS* gene with our previously developed TransCRISTI method [28] (Table 1, Fig 2B). Insertion of the gene

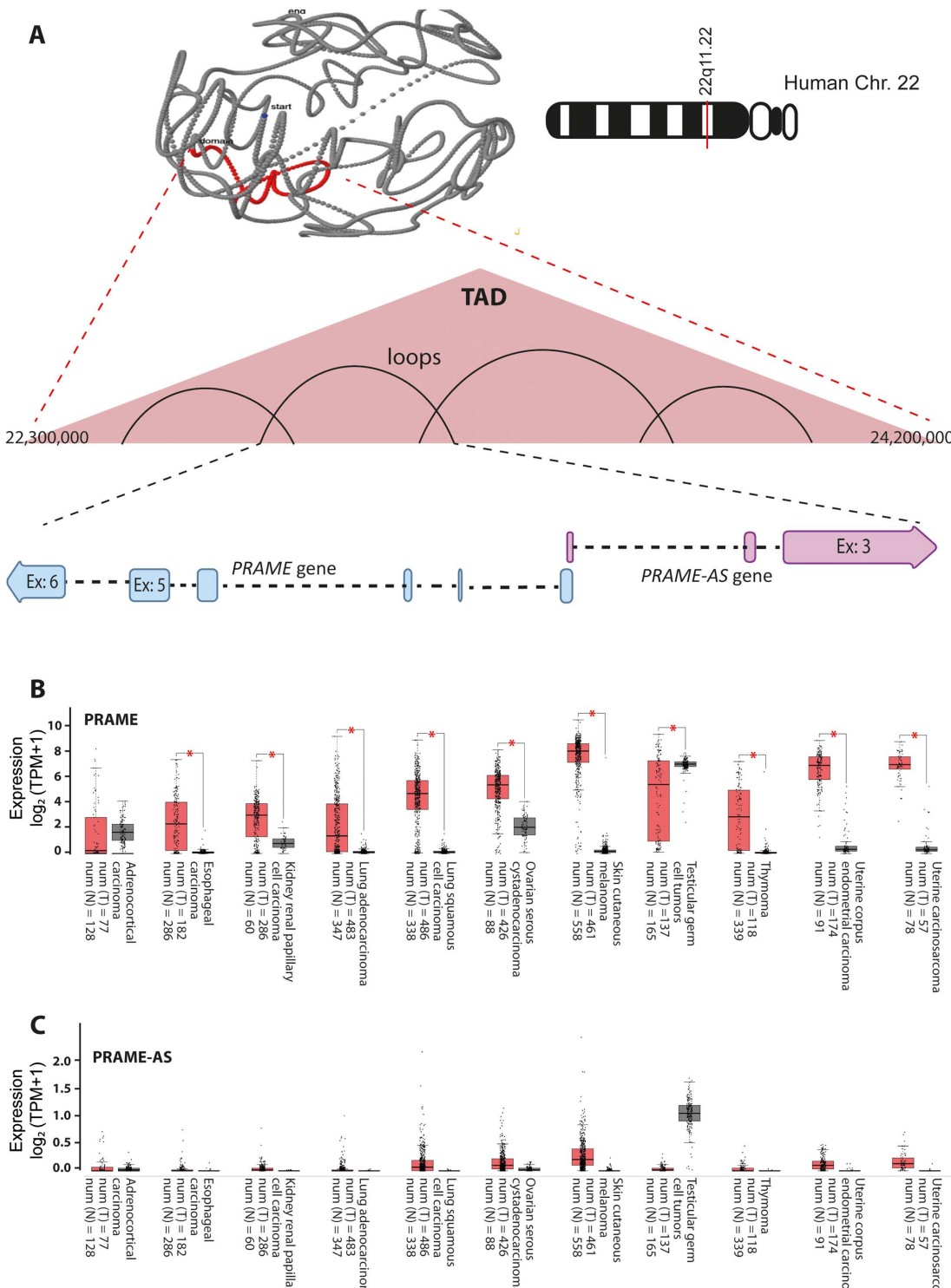

**Fig 1. PRAME and PRAME-AS lncRNA transcript levels differ between cancerous and normal states of solid tissues. A)** The *PRAME* and *PRAME-AS* lncRNA genes are located on the 22q11.22 region of human chromosome 22. These two genes belong to a single topologically associated domain (TAD) in the 3D structure of this chromosome (http://dna.cs.miami.edu/TADKB/browse.php). The median expression values of PRAME **(B)** and PRAME-AS lncRNA transcripts **(C)**, extracted from the GEPIA2 webserver, are shown in different cancers (red boxes) and their corresponding normal tissues (gray boxes) in units of transcript per million (TPM). N: Normal, T: Tumor, Num: Numbers. Statistical differences in both panels of B and C were based on a $p<0.01$. The asterisk (*) shows a $p<0.01$ level of statistical significance.

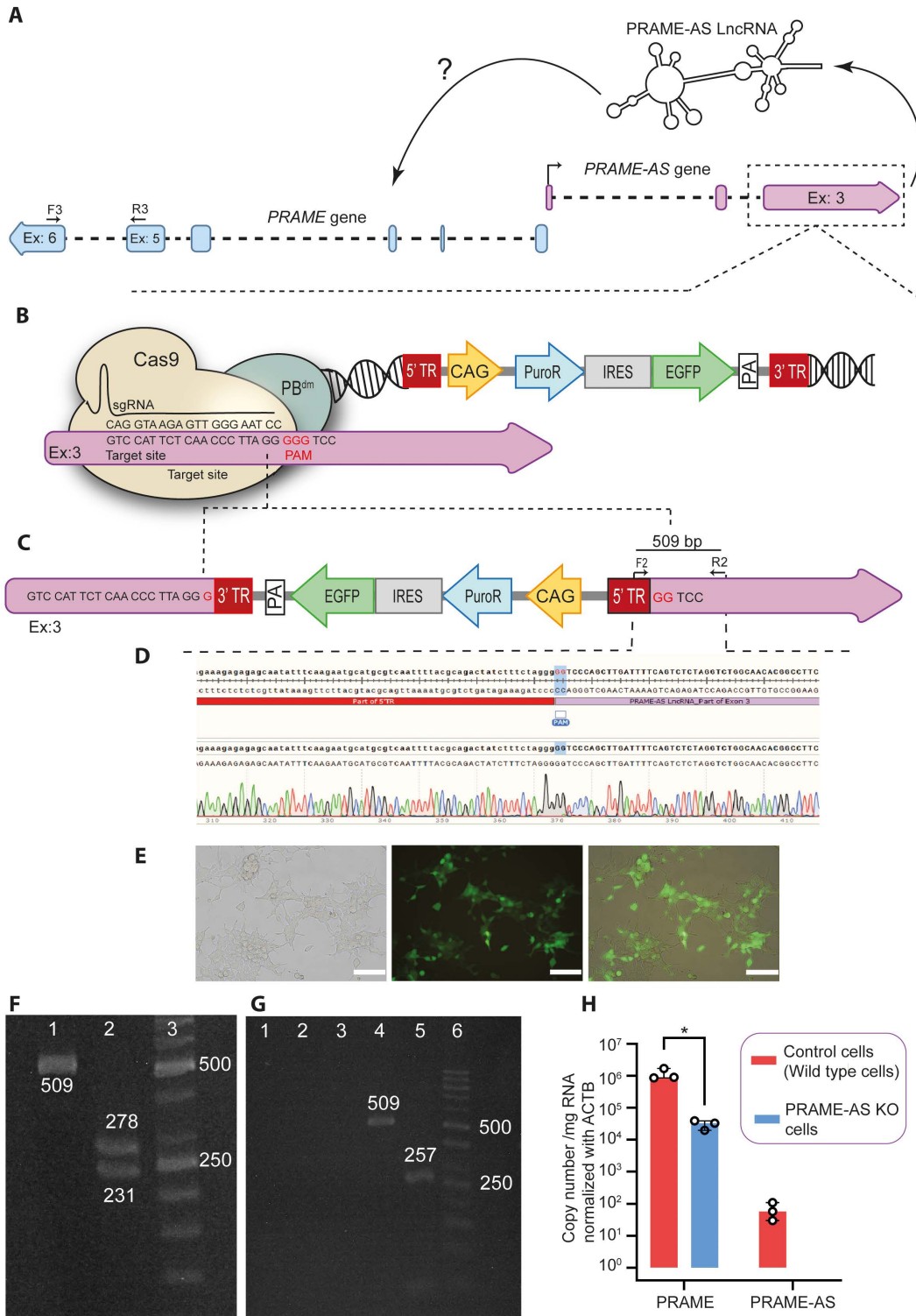

**Fig 2. Ablation of the *PRAME-AS* gene affects PRAME transcript levels. A)** Hypothetical regulation of PRAME by PRAME-AS transcript. **B)** Schematic overview of *PRAME-AS* gene knockout using the TransCRISTI method. **C)** Insertion of the donor cassette in the third exon of the *PRAME-AS* gene. **D)** A Chromatogram of Sanger sequencing reveals the insertion of the donor fragment in the targeted region. **E)** The brightfield, fluorescent, and merged images of the *PRAME-AS* knockout cell line (Scale bar: 100 μm). **F)** Agarose gel electrophoresis analysis of PCR product amplified from the insertion location. Lane 1: The PCR amplification of the third exon of the *PRAME-AS lncRNA* gene after insertion (using F2 and R2 primers) yields a

509 bp product. Lane 2: 278 bp and 231 bp bands after SacI digestion of the 509 bp amplicon. Lane 3: A 50 bp DNA size marker. **G)** Agarose gel electrophoresis analysis of RT-PCR products amplified from the insertion location. Lane 1: NTC (no template control), Lane 2: RT minus (minus-reverse transcriptase control), Lane 3: The RT-PCR amplification of the third exon of the PRAME-AS lncRNA transcripts after insertion (using F2 and R2 primers) is expected to produce a 509 bp product. Lane 4: The PCR amplification of the third exon of the *PRAME-AS lncRNA* gene after insertion (using F2 and R2 primers) yields a 509 bp product as a positive control. Lane 5: The RT-PCR amplicon of ACTB (as a cDNA quality control), Lane 6: A 50 bp DNA size marker. **H)** Fold-change of ACTB-normalized PRAME and PRAME-AS transcripts in the *PRAME-AS* gene knockout cells (n = 3) compared with wild type cells (n = 3). Data from three independent experiments were summarized as median and range and were statistically analyzed using the Mann-Whitney U-test. The asterisk (*) shows a $p = 0.05$ level of statistical significance.

cassette into the PRAME-AS lncRNA genomic region of HEK293T cells (Fig 2C) resulted in the stable expression of *EGFP* and *PAC* (puromycin N-acetyltransferase; PuroR) genes (Fig 2E). The extracted genomic DNA from the enriched EGFP-expressing cells was subjected to PCR (Table 1 and Fig 2C) and gel electrophoresis (Fig 2F) to confirm the correct genomic integration of the knockout fragment. The PCR amplicon was verified by restriction enzyme digestion (Fig 2F) and Sanger sequencing (Fig 2D). We were not able to detect the PRAME-AS transcripts in *PRAME-AS* gene knockout cells (Fig 2G). The RT-qPCR quantification revealed a 37% reduction of PRAME transcript levels in *PRAME-AS* gene knockout cells ($p = 0.05$) (Fig 2H). Thus, PRAME-AS lncRNA can be considered a regulator of PRAME transcript levels.

## The PRAME-AS knockout cells showed a decrease in migration, proliferation, stemness, and viability

In *PRAME-AS* gene knockout cells, the read absorbance of the MTT assay was significantly lower (40.63%) than that in wild-type cells, indicating that the metabolic activity and cell survival are affected ($p < 0.0001$) (Fig 3A).

A colony formation assay was applied to evaluate cell proliferation in *PRAME-AS* gene knockout cells. After nine days of culture, the number of colonies in wild-type cells was approximately twice that observed in *PRAME-AS* gene knockout cells (on average, four colonies versus two colonies per microscopic field with 10x magnification) ($p < 0.0001$). In addition, the average colony diameter in wild-type cells was significantly greater than that observed in *PRAME-AS* gene knockout cells (about 1000 pixels versus 700 pixels) ($p < 0.0001$) (Fig 3B).

We employed a soft agar colony formation assay to evaluate anchorage-independent growth and cellular stemness. After one week of culture in low-melting point agarose, wild-type cells produced more colonies than *PRAME-AS* gene knockout cells ($p < 0.05$). Additionally, the colony diameter in wild-type cells was significantly greater than that in *PRAME-AS* gene knockout cells (average diameter size of 449 pixels versus 391 pixels; $p < 0.0001$) (Fig 3C).

We also assessed the migratory behavior of the *PRAME-AS* gene knockout cells by using a wound healing (scratch) assay. Twenty-four hours post-scratch, the migration of *PRAME-AS* knockout cells was markedly reduced compared to wild-type cells ($p < 0.05$) (Fig 3D).

## Overexpression of MZF1 leads to elevated transcript levels of both PRAME and PRAME-AS lncRNA

The first intron of the *PRAME* gene, which overlaps with the 5′ upstream region of the *PRAME-AS* gene (Fig 4Aa), contains multiple binding sites for the MZF1 transcription factor (Fig 4Ab). Given that MZF1 binding to this intronic region has been shown to upregulate PRAME expression in melanoma cells [23], we investigated its regulatory influence on PRAME-AS lncRNA transcript levels. Using *piggyBac* transposition, we generated a HEK293T cell line that stably expresses MZF1 (Fig 4B). We also generated a control HEK293T cell line that contained the same *piggyBac* transposon without the MZF1 coding sequence (Fig 4C). The overexpression of the MZF1 protein was confirmed by immunocytochemistry (Fig 4B and S8 Fig). In response to MZF1 overexpression, transcript levels of PRAME and PRAME-AS lncRNA increased approximately two- and three-fold, respectively, compared to the control cell line ($p < 0.05$) (Fig 4D and S9 Fig).

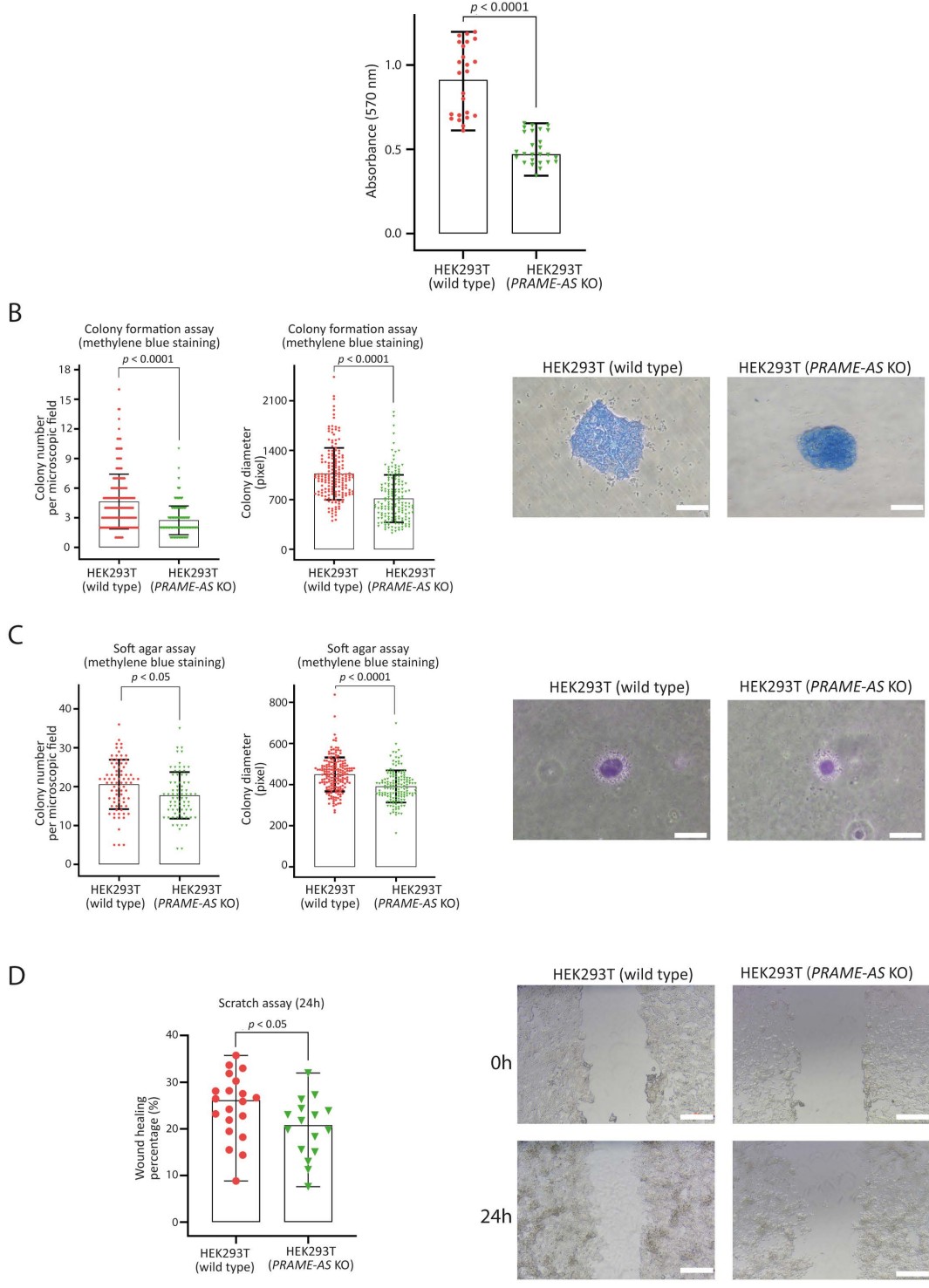

**Fig 3. The PRAME-AS knockout cells showed a decrease in migration, proliferation, stemness, and viability. A)** MTT assay to evaluate cell viability of HEK293T wild-type (n = 24) and *PRAME-AS* gene knockout cells (n = 26) forty-eight hours after cell seeding (read absorbance on 570 nm). Data are shown as median and range. **B)** Findings of colony formation assay for evaluation of proliferation in HEK293T wild-type (Number of analyzed microscopic fields: n = 197 for colony number evaluation, and n = 175 for colony diameter evaluation) and *PRAME-AS* gene knockout cells (Number of

analyzed microscopic fields: n = 226 for colony number evaluation, and n = 165 for colony diameter evaluation) after nine days of culture. The two rightmost panels display representative images of colonies. Data are shown as mean and SD (Scale bar: 100 µm). **C)** Soft agar colony formation assay for evaluation of anchorage-independent growth and cell stemness in HEK293T wild-type (Number of analyzed microscopic fields: n = 80 for colony number evaluation, and n = 181 for colony diameter evaluation) and *PRAME-AS* gene knockout cells (Number of analyzed microscopic fields: n = 75 for colony number evaluation, and n = 145 for colony diameter evaluation) after one week of culture. The two rightmost panels display representative images of colonies. Data are shown as mean and SD (Scale bar: 100 µm). **D)** Wound-healing assay for evaluating the migratory potential of HEK293T wild-type (n = 20) and *PRAME-AS* gene knockout cells (n = 16) after twenty-four hours of scratching. Data are shown as median and range. The scale bar indicates a length of 500 µm. The size of colonies and scratch areas was measured using ImageJ software. All experiments were performed with at least three technical and two biological replicates. The statistical differences between the wild-type and *PRAME-AS* gene knockout cells were analyzed by the Mann-Whitney U-test for nonparametric data (MTT and wound healing assays) and the Welch correction test for parametric data (soft agar and colony formation assays).

### CRISPR/dCas9-DNMT3A-mediated DNA methylation of a regulatory region in the PRAME locus affects PRAME and PRAME-AS lncRNA transcripts

Based on a previous report, the hypomethylation of a CpG island in the first intron of *PRAME* affects its expression in human malignancies [22]. Since the first intron of the *PRAME* gene overlaps with the 5′ upstream regulatory region of the *PRAME-AS* gene (Fig 5Aa), we identified the exact positions of CpGs in this region (Fig 5Ab). Then, using the dCas9-DNMT3A tool and the design of targeting sgRNAs 25−35 base pairs downstream of the PAM sequence as recommended by Vojta et al. [30] (Fig 5Ab), we manipulated the methylation status of these CpGs. The HEK293T cells were transiently transfected by control and sgRNA-encoding plasmids (Fig 5B and 5C). Sanger sequencing of the amplicon encompassing the target region from transfected cells (with pHD_71666–1: dCas9-DNMT3A + sgRNA#1 and pHD_71666–2: dCas9-DNMT3A + sgRNA#2) confirmed hypermethylation of the CpG dinucleotides compared with untransfected cells, which exhibited only baseline methylation levels (Fig 5D). RT-qPCR analysis of RNA from dCas9-DNMT3A-treated cells revealed that PRAME and PRAME-AS transcript levels were reduced by approximately 50% (p < 0.05) and 18.5%, respectively, compared to control cells (Fig 5E).

### The PRAME locus regulatory region regulates transcription of a reporter gene only in the PRAME direction

*PRAME and PRAME-AS* genes are located in a head-to-head configuration with overlapping 5′ ends (Fig 6A). This feature, along with two CpG islands, numerous SP1 binding sites in addition to other signs of active transcription, such as H3K27ac (open chromatin signals) and enhancer sequence, might indicate the presence of a bidirectional promoter, working for both genes (Fig 6A). The HEK293T cells were transiently transfected with the EGFP reporter constructs, containing a 1361-bp sequence (PRAME locus regulatory region) as promoter (containing two annotated TSS; [22]) in two opposite directions (Fig 6B). We observed reporter activity only in those cells that had the construct in the PRAME direction (Fig 6C–6E). Flow cytometry analysis also displayed that the percentage of EGFP-expressing cells in the PRAME direction (14.6%) was more than that in the promoterless vector (4.81%) (Fig 6F and 6G). However, the PRAME locus regulatory region in the opposite direction could not induce significant expression of the EGFP protein (5.47%) (Fig 6F and 6G). Based on our EGFP reporter assay, we conclude that the regulatory region of the PRAME locus—without PRAME transcription—is insufficient to drive transcription of PRAME-AS in the antisense direction.

## Discussion

In this study, utilizing a knockout strategy, we demonstrate that the PRAME-AS lncRNA functions as a regulator of PRAME transcript levels (Fig 2). In knockout cells, we observed a significant reduction in cell viability, proliferation rate, anchorage-independent growth, and migration capacity (Fig 3). We also show that the MZF1 transcription factor influences the expression of the *PRAME-AS* gene (Fig 4). Additionally, our experiments reveal that the methylation status of a CpG island, located in a regulatory region shared with the *PRAME* gene, affects the expression of both genes (Fig 5).

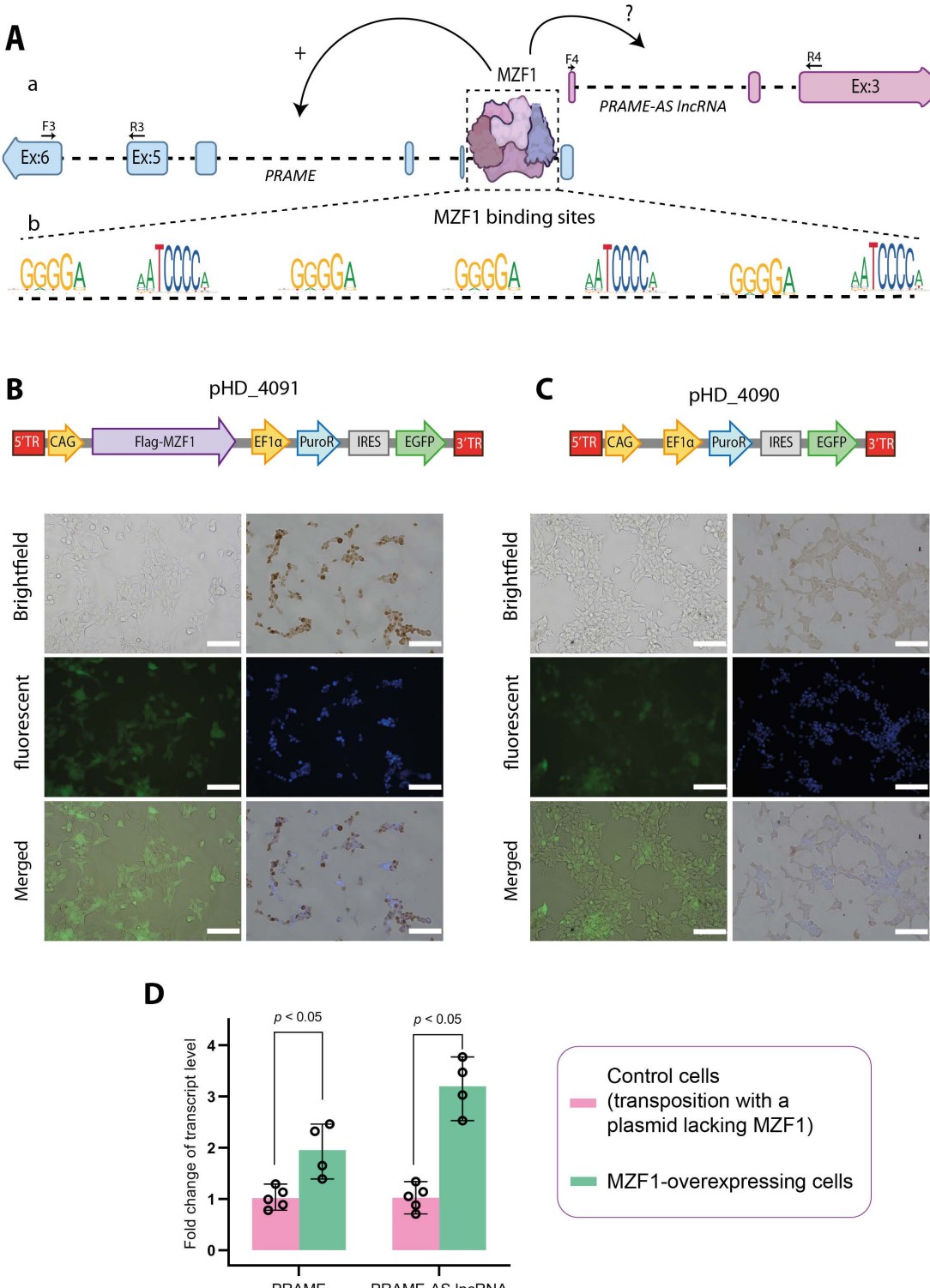

**Fig 4. Overexpression of MZF1 leads to elevated transcript levels of both PRAME and PRAME-AS lncRNA.** Aa) A schematic illustration of the hypothetical regulation of PRAME and PRAME-AS transcription by MZF1 transcription factor. Ab) Potential MZF1 binding sites on the PRAME genomic region based on JASPAR (http://jaspar.genereg.net) and GPminer (http://gpminer.mbc.nctu.edu.tw) databases. **B)** A *piggy*Bac transposition was used to create HEK293T cells overexpressing FLAG-MZF1 under the control of the CAG promoter and PuroR-IRES-EGFP under the control of the EF1α

promoter. Panels show brightfield, fluorescent, DAB-stained, and DAPI-stained images of the FLAG-MZF1 overexpressing HEK293T cells. **C)** Control cells were generated via transposition using a transposon lacking the FLAG-MZF1 coding sequence and containing only the PuroR-IRES-EGFP cassette driven by the EF1α promoter. Panels show brightfield, fluorescent, DAB-stained, and DAPI-stained images. **D)** PRAME and PRAME-AS transcript levels were elevated in the FLAG-MZF1-overexpressing cell line compared to the control cell line (n = 5 in control cell line and n = 4 in the MZF1-overexpressing cell line). Transcript levels of PRAME and PRAME-AS were quantified by RT-qPCR from three biological replicates. The reported levels of these transcripts were normalized to the *ACTB* reference gene and reported as median with range. Statistical differences between the control and MZF1-overexpressing cell lines were analyzed using the Mann-Whitney U-test.

We also investigated the ability of the PRAME locus regulatory region to drive the transcription of PRAME-AS lncRNA (Fig 6), and concluded that this regulatory region—without PRAME transcription—is insufficient to drive transcription of PRAME-AS in the antisense direction.

Several studies have reported that antisense lncRNAs regulate the expression of their corresponding sense protein-coding genes. MUC5B-AS1 via a mutual direct binding stabilizes MUC5B mRNA, giving rise to metastasis and poor survival in lung adenocarcinoma [38–40]. Similarly, the PXN-AS1-L isoform—containing exon 4—directly interacts with the 3′ untranslated region (3′UTR) of PXN mRNA, inhibiting its degradation by the miR-24–AGO2 complex and consequently upregulating PXN expression in hepatocellular carcinoma [41]. Oncogenic FOXP4-AS1 lncRNA, by absorbing miR-3184-5p, indirectly leads to the upregulation of FOXP4 mRNA in prostate cancer [42]. The SATB2-AS1 recruits WDR5 and GADD45A to the SATB2 promoter and catalyzes DNA demethylation and histone 3 lysine 4 trimethylation to prohibit colorectal cancer cells' progression [43]. This is the first study to investigate PRAME-AS and its effects on PRAME transcript levels (Fig 2). Further investigation is required to elucidate the precise mechanisms by which PRAME-AS regulates PRAME expression.

As previously indicated, there is strong evidence supporting the role of PRAME as a key contributor to cellular malignancy across various cancer types. For example, in triple-negative breast cancer (TNBC), PRAME overexpression induces the expression of 11 EMT (epithelial- to-mesenchymal transition)-related genes and leads to deregulation of Notch and Wnt signaling pathways, inhibition of apoptosis, and degradation of basement membranes in blood vessels [8]. Moreover, PRAME is elevated in hepatocellular carcinoma and prohibits apoptosis mediated by p53 and Bcl2 [9]. High level of PRAME has a strong positive correlation with the EMT-related transcription factors SNAI1 and TWIST, while normal liver regulators, including FOXA2 and AR, are negatively correlated with PRAME expression [26]. PRAME promotes the proliferation of osteosarcoma cells [10], while in Chronic myelogenous leukemia, in the presence of RA and PcG proteins, including EZH2, it binds to retinoic acid receptor (RAR) to block Caspase 3, TRAIL, and P21 expression, leading to proliferation and apoptosis arrest [14,15]. Another study in seminomas indicates that the overexpression of PRAME and its interaction with Lin28A support pluripotency networks and expression of reprogramming markers, such as DPPA3, Nanog, Oct3/4, ZSCAN10, and PRDM14 [20]. Two distinct research reports on non-small cell lung cancer (NSCLC) patients of Asia (South Korea, Singapore, Taiwan, and Thailand) showed a higher rate of PRAME transcripts in squamous cell carcinoma (SCC), and in smokers compared with non-smokers [16,17]. A recent study on cervical cancer indicates PRAME's involvement in proliferation, invasion, migration, reduction of apoptotic cells, and facilitating G0/G1 cell-cycle by the Wnt/β-Catenin pathway [21]. Another study supports the role of PRAME in proliferation, invasion, migratory potential, and EMT in LSCC, and the growth of tumors in response to PI3K/AKT/mTOR pathway activation [27]. These reports highlight the involvement of diverse genetic and epigenetic factors, regulatory networks, and pathways downstream of PRAME that influence cellular functions such as proliferation, viability, and stemness. In this study, we expand the players to add the PRAME-AS lncRNA to this regulatory network. However, since in this study, we manipulated only the *PRAME-AS* gene, we cannot expect to observe considerable changes in all kinds of cellular functions (Fig 3). For example, in the PRAME-AS knockout cells, MZF1, which is a positive regulator of PRAME, is still expressed and will continue to play its regulatory role.

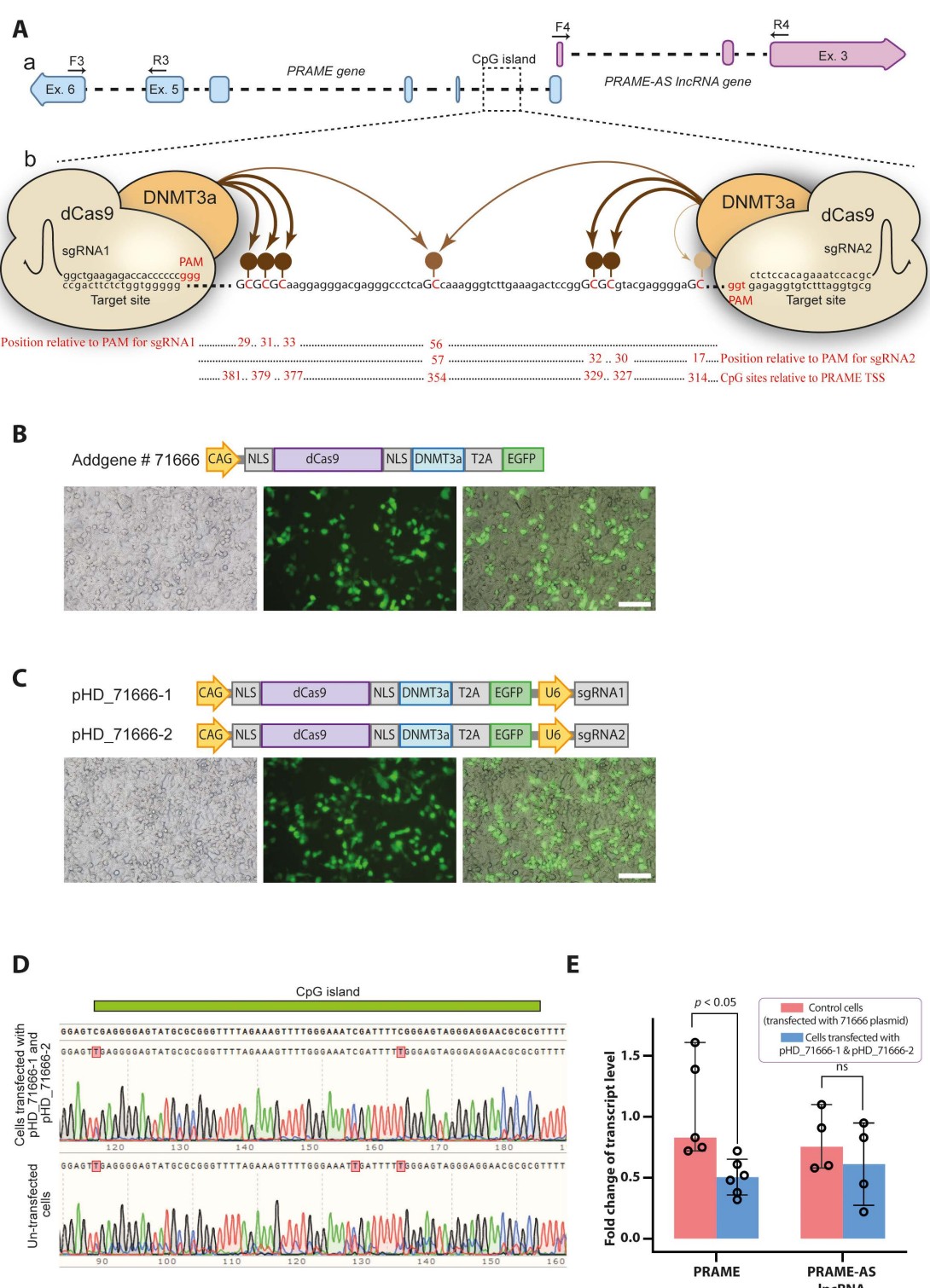

**Fig 5. Hypermethylation of a regulatory region in the PRAME locus affects PRAME and PRAME-AS lncRNA transcripts.** Aa) Graphic representation of a regulatory region in the PRAME locus containing a CpG island, which may be involved in the regulation of transcription of *PRAME-AS* gene. Ab) Schematic illustration of dCas9-DNMT3A effectors targeting a CpG island using two specific sgRNAs that bind 25–35 bp downstream (the effective range for a fully methylated state) of the PAM sequences [30]. The position of CpG dinucleotides is shown relative to the nearest sgRNA and to the PRAME transcriptional start site. The thickness and higher color intensity of the brown arrows correspond to the higher DNMT3A activity. **B)** Cells

transfected with the construct encoding dCas9-DNMT3A-T2A-EGFP (Addgene #71666) (Scale bar: 100 μm). **C)** Cells transfected with constructs encoding dCas9-DNMT3A-T2A-EGFP and sgRNA#1 (pHD_71666-1) and dCas9-DNMT3A-T2A-EGFP and sgRNA#2 (pHD_71666-2). (Scale bar: 100 μm). **D)** Sanger sequencing of the bisulfite-treated genomic DNA from cells transfected with pHD_71666-1 (dCas9-DNMT3A + sgRNA # 1) and pHD_71666-2 (dCas9-DNMT3A + sgRNA # 2) compared with un-transfected cells with baseline methylation levels. **E)** RT-qPCR results after normalization by *ACTB* reference gene reveal a drop of PRAME (n = 5 in control cell line and n = 6 in dCas9-DNMT3A- treated cells) and PRAME-AS lncRNA (n = 4) transcripts after hypermethylation. The data was gathered from three biological replicates, represented as median and range, and statistically analyzed by Mann-Whitney U-test.

MZF1 promotes various types of cancers by targeting key regulators of tumor progression, including c-Myc, AXL, IGF1R, and PKCα [44–52]. Lee et al. reported that MZF1 is localized to the first intron of PRAME,and there is a positive relationship between MZF1 and PRAME expression [23]. In this study, our initial *in silico* observations that MZF1 binding sites possibly overlap with PRAME-AS promoter prompted us to identify whether MZF1 acts as a coregulator of PRAME and PRAME-AS expression. We found that PRAME-AS and PRAME transcripts were increased in MZF1 overexpressed cell lines (Fig 4). On the other hand, DNA demethylation has been shown to facilitate PRAME expression in various cancers [22,53–57]. Similarly, hypomethylation, in cooperation with MZF1, enhances PRAME expression in melanoma [23]. As expected, our results showed a 50% reduction in PRAME transcript levels in dCas9-DNMT3A-treated cells. Correspondingly, PRAME-AS expression decreased by 18.5% in these cells (Fig 5).

Several studies have reported that two sense and antisense genes in a head-to-head arrangement may share regulatory regions between their transcription start sites [58–63]. This shared intergenic region is characterized by an enrichment of CpG content and numerous SP1 binding sites [64,65]. It is speculated that these features may assist the co-regulation or co-function of genes [65]. However, our reporter assay that includes the PRAME locus regulatory region did not reveal a considerable promoter function in the PRAME-AS direction (Fig 6). In support of our findings, a recent study reported that a ~ 300-nucleotide common regulatory region with signatures of a bidirectional promoter for the genes *LINC00882* and *DUBR*, lost its promoter activity when upstream regions were also cloned next to this common region [66]. Thus, one possible explanation for our findings is that the promoter activity of the defined region may have been negatively affected by additional upstream sequences.

In conclusion, our study elucidates the regulatory role of the antisense lncRNA PRAME-AS in controlling PRAME transcript levels. We further demonstrate that MZF1 and a CpG island contribute significantly to the modulation of both PRAME-AS and PRAME expression. These findings suggest that the regulatory axis involving PRAME-AS, MZF1, and epigenetic elements such as CpG islands plays a pivotal role in PRAME regulation in cancers, highlighting PRAME-AS as a potential key contributor to tumor processes.

## Supporting information

**S1 Fig. Comparison of PRAME protein expression between tumor and normal tissues.** Breast Cancer and Glioblastoma do not have normal samples. The data was derived from LinkedOmicsKB web tool (https://kb.linkedomics.org/). (PDF)

**S2 Fig. Map of the pHD_4091 plasmid and the confirmation of the FLAG-MZF1 cloning by Sanger sequencing.** (PDF)

**S3 Fig. Map of the pHD_4016–1 plasmid, and the insertion of the PRAME locus regulatory region in the PRAME direction verified by Sanger sequencing.** (PDF)

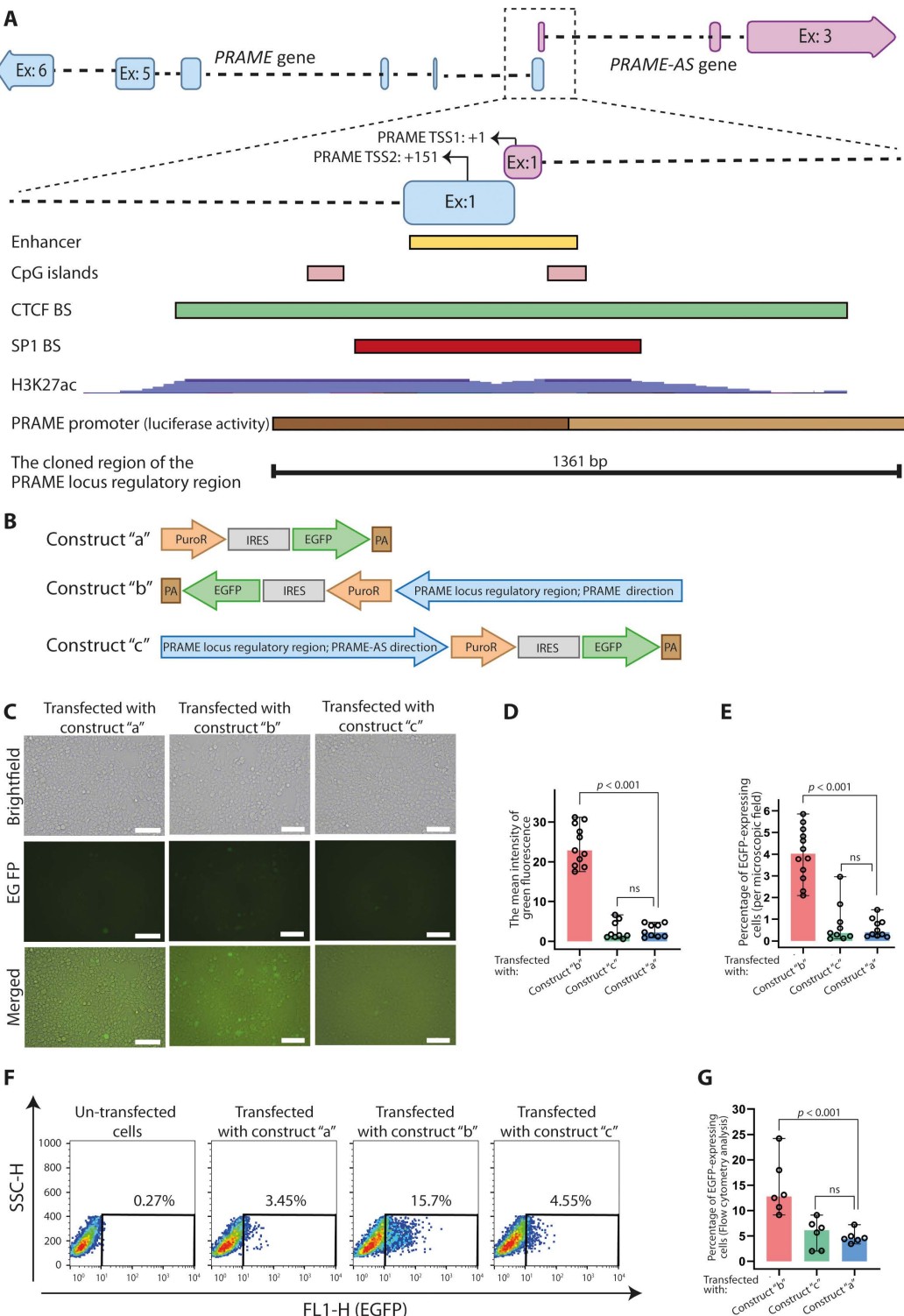

**Fig 6. The PRAME locus regulatory region regulates transcription of a reporter gene only in the PRAME direction. A)** Genomic location of *PRAME* and *PRAME-AS* genes along with regulatory elements near transcriptional start sites. Regulatory features of this region are shown as the yellow rectangle (enhancer sequence; NCBI database, Gene ID: 108491837), two pink rectangles (CpG islands [22]), the green rectangle (CTCF-binding sites; UCSC Genome Browser), the red rectangle (SP1-binding sites; UCSC Genome Browser), blue histogram (H3K27ac; UCSC Genome Browser), brown

rectangle (predicted PRAME promoter region based on luciferase activity [22], dark brown; high luciferase activity, light brown; moderate luciferase activity), and black line (representation of 1361-bp DNA fragment cloned into pHD_4016 vector). **B)** Schematics of pHD_4016 vector without a promoter (pHD_4016-3; construct "a"), with PRAME locus regulatory region in the PRAME orientation (pHD_4016-1; construct "b"), and with PRAME locus regulatory region in the PRAME-AS orientation (pHD_4016-2; construct "c"). **(C)** Brightfield, fluorescent, and merged images of HEK293T cells 48h after transient transfection of construct "a", construct "b", and construct "c", (Scale bar: 100 μm). **D)** ImageJ analysis of the mean intensity of green fluorescence (Number of analyzed images: n = 11 in cells transfected with construct "b", n = 9 in cells transfected with construct "c" and cells transfected with construct "a"). **E)** The percentage of EGFP-expressing cells per microscopic field of the transfected cells (Number of analyzed microscopic fields: n = 12 in cells transfected with construct "b", n = 9 in cells transfected with construct "c" and n = 10 in cells transfected with construct "a"). **F)** Representative flow cytometry results to assay EGFP expression. **G)** The percentage of EGFP expression in the transfected HEK293T cells is illustrated as median and range (Number of samples in each group: 6). The experiment is designed and done in three biological replicates, each in duplicate. Significant differences among experimental groups were analyzed using the Mann-Whitney U-test and represented as median and range.

**S4 Fig. Map of the pHD_4016–2 plasmid, and the insertion of the PRAME locus regulatory region in the PRAME-AS direction confirmed by Sanger sequencing.**
(PDF)

**S5 Fig. (A) Melting curves, (B) gel electrophoresis analysis of PCR products, and (C) standard curves for ACTB, PRAME-AS, and PRAME.** PCR amplicons for ACTB (lane 2; 257 bp), PRAME-AS (lane 3; 312 bp), and PRAME (lane 4; 212 bp). Lane 1; 50 bp DNA size marker.
(PDF)

**S6 Fig. Determination of the proper amount of serum in the DMEM medium for the scratch assay.** The color of DMEM medium (A) and wound healing closure (B) in HEK293T cells exposed to different percentages of serum between two time points 0h and 24h after scratch (scale bar: 500 μm).
(PDF)

**S7 Fig. Comparative expression of PRAME and PRAME-AS lncRNA in normal and cancerous tissues.** Dot plots compare the median transcript levels of PRAME (A) and PRAME-AS lncRNA (B) in tumor (red dots) and normal samples (green dots) from the TCGA and the GTEx databases, respectively. N: Normal, T: Tumor, n: Numbers. ACC; Adrenocortical carcinoma, BLCA; Bladder Urothelial Carcinoma, BRCA; Breast invasive carcinoma, CESC; Cervical squamous cell carcinoma and endocervical adenocarcinoma, CHOL; Cholangio carcinoma, COAD; Colon adenocarcinoma, DLBC; Lymphoid Neoplasm Diffuse Large B-cell Lymphoma, ESCA; Esophageal carcinoma, GBM; Glioblastoma multiforme, HNSC; Head and Neck squamous cell carcinoma, KICH; Kidney Chromophobe, KIRC; Kidney renal clear cell carcinoma, KIRP; Kidney renal papillary cell carcinoma, LGG; Brain Lower Grade Glioma, LIHC; Liver hepatocellular carcinoma, LUAD; Lung adenocarcinoma, LUSC; Lung squamous cell carcinoma, MESO; Mesothelioma, OV; Ovarian serous cystadenocarcinoma, PAAD; Pancreatic adenocarcinoma, PCPG; Pheochromocytoma and Paraganglioma, PRAD; Prostate adenocarcinoma, READ; Rectum adenocarcinoma, SARC; Sarcoma, SKCM; Skin Cutaneous Melanoma, STAD; Stomach adenocarcinoma, TGCT; Testicular Germ Cell Tumors, THCA; Thyroid carcinoma, THYM; Thymoma, UCEC; Uterine Corpus Endometrial Carcinoma, UCS; Uterine Carcinosarcoma, UVM; Uveal Melanoma.
(PDF)

**S8 Fig. Validation of the MZF1 protein expression in the MZF1-overexpressing cells.** (A) Immunocytochemical detection of FLAG-MZF1 in the nucleus of MZF1-overexpressing cells (brown nuclei) (B) and the negative control cells (colorless nuclei) (Scale bar: 50 μm).
(PDF)

**S9 Fig. Up-regulation of PRAME and PRAME-AS lncRNA transcript levels in the MZF1 overexpressing cells relative to the control cells.** The RNA copy numbers were quantified by RT-qPCR in three biological replicates. The RNA

copy numbers for PRAME and PRAME-AS were normalized by the *ACTB* reference gene and reported as median with range. Statistical analysis was done using the Mann-Whitney U-test.
(PDF)

**S1 Raw Images. Agarose gel electrophoresis analysis of PCR products in this study.** A) The insertion location amplified by PCR. Lane 1: The PCR amplification of the third exon of the *PRAME-AS lncRNA* gene after insertion (using F2 and R2 primers) yields a 509 bp product. Lane 2: Restriction enzyme digestion of the 509 bp amplicon to confirm its sequence; 231 and 278 bp bands upon SacI digestion. Lane 3: A 50 bp DNA size marker (This image was used in Fig 2F). B) The RT-PCR products amplified from the insertion location. Lane 1: No template control (NTC), Lane 2: RT minus (minus-reverse transcriptase control), Lane 3: The RT-PCR amplification of the third exon of the PRAME-AS lncRNA transcripts after insertion (using F2 and R2 primers), expected to produce a 509 bp product. Lane 4: The PCR amplification of the third exon of the *PRAME-AS lncRNA* gene after insertion (using F2 and R2 primers) yields a 509 bp product as a positive control. Lane 5: The RT-PCR amplicon of ACTB (as a quality control for cDNA), Lane 6: A 50 bp DNA size marker (This image was used in Fig 2G). C) PCR amplicons for ACTB (lane 2; 257 bp), PRAME-AS (lane 3; 312 bp), and PRAME (lane 4; 212 bp). Lane 1; A 50 bp DNA size marker (This image was used in S5 Fig). In all cases, after electrophoresis, the agarose gels were stained by ethidium bromide. In order to visualize DNA bands, the stained gels were exposed to ultraviolet (UV) light.
(PDF)

**S1 File. The 5′ to 3′ sequence of the cloned region of the PRAME locus regulatory region.**
(PDF)

**S1 Table. Details of HEK293T transfection for different applications of gene knockout, gene overexpression, CpG methylation, and PRAME locus regulatory region assay.**
(PDF)

## Acknowledgments

We would like to kindly thank Prof. Soo-Jong Um and Nackhyoung Kim for Flag/Myc-tagged hMZF1 plasmids, and Prof. Hans Peter Saluz and Dr. Tino Schenk for their fruitful and constitutive assistance in mapping CpG islands based on new annotations. We are grateful to Dr. Saeideh Nakhaeirad for the anti-mouse IgG, HRP-linked antibody.

## Author contributions

**Conceptualization:** Zahra Hosseininia, Hesam Dehghani.

**Data curation:** Zahra Hosseininia, Hesam Dehghani.

**Formal analysis:** Zahra Hosseininia, Hesam Dehghani.

**Funding acquisition:** Hesam Dehghani.

**Investigation:** Zahra Hosseininia, Hesam Dehghani.

**Methodology:** Zahra Hosseininia, Hesam Dehghani.

**Project administration:** Hesam Dehghani.

**Resources:** Hesam Dehghani.

**Software:** Zahra Hosseininia, Hesam Dehghani.

**Supervision:** Hesam Dehghani.

**Validation:** Hesam Dehghani.

**Visualization:** Zahra Hosseininia, Hesam Dehghani.

**Writing – original draft:** Zahra Hosseininia.

**Writing – review & editing:** Hesam Dehghani.

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
