## [Decision Letter · Decision Letter 0]

7 Mar 2025

PONE-D-25-03086PRAME-AS lncRNA is regulated by MZF1 and is involved in the expression of PRAME transcripts and cell stemnessPLOS ONE

Dear Dr. Dehghani,

Thank you for submitting your manuscript to PLOS ONE. After careful consideration, we feel that it has merit but does not fully meet PLOS ONE’s publication criteria as it currently stands. Therefore, we invite you to submit a revised version of the manuscript that addresses the points raised during the review process.

 **Your manuscript was reviewed by four experts. While they found it potentially interesting, they raised some concerns. One of the major issues is that experimental evidence is insufficient to support your conclusions. Please revise it according to their suggestions. Some additional experiments would be necessary.**

We look forward to receiving your revised manuscript.

Kind regards,

Hodaka Fujii, M.D., Ph.D.

Academic Editor

PLOS ONE

**Journal Requirements:**

1. When submitting your revision, we need you to address these additional requirements. Please ensure that your manuscript meets PLOS ONE's style requirements, including those for file naming. The PLOS ONE style templates can be found at https://journals.plos.org/plosone/s/file?id=wjVg/PLOSOne_formatting_sample_main_body.pdf and https://journals.plos.org/plosone/s/file?id=ba62/PLOSOne_formatting_sample_title_authors_affiliations.pdf 2. Thank you for stating in your Funding Statement:  This work was supported by a Ferdowsi University of Mashhad (http://en.um.ac.ir/) grant to HD (No. 53395). The funders had no role in study design, data collection and analysis, decision to publish, or preparation of the manuscript. Please provide an amended statement that declares *all* the funding or sources of support (whether external or internal to your organization) received during this study, as detailed online in our guide for authors at http://journals.plos.org/plosone/s/submit-now.  Please also include the statement “There was no additional external funding received for this study.” in your updated Funding Statement. Please include your amended Funding Statement within your cover letter. We will change the online submission form on your behalf. 3. Please amend either the abstract on the online submission form (via Edit Submission) or the abstract in the manuscript so that they are identical. 4. PLOS ONE now requires that authors provide the original uncropped and unadjusted images underlying all blot or gel results reported in a submission’s figures or Supporting Information files. This policy and the journal’s other requirements for blot/gel reporting and figure preparation are described in detail at https://journals.plos.org/plosone/s/figures#loc-blot-and-gel-reporting-requirements and https://journals.plos.org/plosone/s/figures#loc-preparing-figures-from-image-files. When you submit your revised manuscript, please ensure that your figures adhere fully to these guidelines and provide the original underlying images for all blot or gel data reported in your submission. See the following link for instructions on providing the original image data: https://journals.plos.org/plosone/s/figures#loc-original-images-for-blots-and-gels.   In your cover letter, please note whether your blot/gel image data are in Supporting Information or posted at a public data repository, provide the repository URL if relevant, and provide specific details as to which raw blot/gel images, if any, are not available. Email us at plosone@plos.org if you have any questions.

Reviewers' comments:

Reviewer's Responses to Questions

**Comments to the Author**

1. Is the manuscript technically sound, and do the data support the conclusions?

Reviewer #1: Partly

Reviewer #2: Partly

Reviewer #3: Yes

Reviewer #4: Partly

2. Has the statistical analysis been performed appropriately and rigorously? 

Reviewer #1: Yes

Reviewer #2: Yes

Reviewer #3: Yes

Reviewer #4: Yes

3. Have the authors made all data underlying the findings in their manuscript fully available?

Reviewer #1: Yes

Reviewer #2: No

Reviewer #3: Yes

Reviewer #4: Yes

4. Is the manuscript presented in an intelligible fashion and written in standard English?

Reviewer #1: No

Reviewer #2: Yes

Reviewer #3: Yes

Reviewer #4: Yes

5. Review Comments to the Author

**Reviewer #1: ** This study demonstrates that a lncRNA (PRAME-AS) can regulate its homologous coding gene PRAME. The authors identify MZF1 and hypermethylation of the upstream regulatory region as regulators of this lncRNA's transcription, and ultimately reveal that the PRAME promoter functions unidirectionally in regulating PRAME. While this discovery is novel, the main findings require further experimental validation to strengthen their impact.

Main concerns:

1. The title emphasizes the lncRNA and expression of PRAME, but the narrative lacks cohesion: The authors first identify the lncRNA, after knock out, the PRME expression is reduced, how does LncRNA regulate PRME? without explore further, the author turned to check cell proliferation, migration, viability, and anchorage-independent growth, then they turn to the explore its upstream regulatory factors (MZF1 and hypermethylation), and later shift to analyzing bidirectional promoter activity, Please ensure that all sections are connected naturally and smoothly.

2. The abstract uses non-definitive phrases ("lncRNA MIGHT regulate PRAME", "MZF1 MAY affect both genes"). If the results are reliable, replace tentative terms ("might/may") with stronger language (e.g., "demonstrates" or "mediates") to enhance scientific rigor.

3. PRAME-AS lncRNA transcripts show no significant differential expression between normal and tumor tissues. This raises concerns about its functional role in tumor progression. what drives authors continues to explore this LncRNA in this situation?

4. The study attributes PRAME-AS-mediated phenotypes (proliferation, migration, etc.) to lncRNA activity, but it remains unclear whether these effects are PRAME-dependent or not. To confirm that the observed phenotypes are specifically regulated by PRAME-AS (and not solely by PRAME), include PRAME knockdown/knockout controls in functional assays.

Minor points

1. Figure 1: Include PRAME protein expression to complement transcript-level findings. This could be added as supplementary data.

2. Figure 2G Provide PRAME protein expression results following PRAME-AS knockout to validate transcriptional regulation at the protein level.

3. Figure 4 Include Western blot data to confirm successful MZF1 overexpression.

4. Figure 6C Clearly the cell number of AS direction is much less than the other two.

**Reviewer #2: ** In this paper, titled “PRAME-AS lncRNA is regulated by MZF1 and is involved in the expression of PRAME transcripts and cell stemness”, the authors have undertaken a characterization of a previously unstudied lncRNA originating antisense from tumor antigen PRAME (PRAME-AS). Regulation of PRAME-AS has been analyzed via interrogation of known expression datasets, transcription factor overexpression, the efficacy of its promoter region and the extent of CpG methylation surrounding it. The function of PRAME-AS was assessed via Cas9-mediated ablation with downstream impacts on PRAME expression, proliferation and cell migration.

It is remarkable that for a gene with long known implications in tumorigenesis that the antisense lncRNA expressed at this locus has so far gone uncharacterized and so the authors should be commended for tackling this as a research topic. The method of PRAME-AS KO so as not to alter promoter activity of PRAME is elegant, however there are several concerns regarding the veracity of the conclusions until further clarity is reached.

Major Comments:

Figure 2

1) What is the expression level of PRAME-AS in HEK293T cells and how does it compare to PRAME expression approximately? Low expression levels is not necessarily an impediment to study as lncRNAs are commonly lowly expressed, but it should be made clear relative to housekeeping gene.

2) What is the extent of PRAME-AS decrease after integration of the donor fragment? Preferably shown with primers both upstream (Exon 2) and downstream (Exon 3) of the integrant.

3) There is concern that inserting a construct inversely to PRAME-AS but on the same strand and upstream as PRAME may lead to interference with PRAME transcription if there is transcription readthrough of the construct.

a) Does the sequence have (a) polyadenylation signal(s) to ensure transcription termination of the integrant?

b) Can it be shown that there is no transcription readthrough downstream of the 3’TR region?

4) Figure 2F: The presence of the Puro-IRES-GFP integrant is demonstrated, however it isn’t clear or explicitly stated whether this is integrated into or one both alleles. Can this be shown or clarified?

Figure 3

1) It would be ideal to perform a rescue experiment of overexpressing PRAME in the PRAME-AS KO cells to determine whether those phenotypes are due to the ~34% loss of PRAME.

Figure 4

1) Related to Figure 2, the fold change of PRAME and PRAME-AS can be seen but how do they compare to each other? It would be very interesting to see similar fold increases when overexpressing MZF1 if there were a large basal difference between PRAME and PRAME-AS.

Figure 5

1) If PRAME-AS expression levels are already low, and methylation levels high, then any dCas9 methylation will likely not alter PRAME-AS expression levels further. Can the level of methylation of each of the CpGs analyzed with bisulfite sequencing be quantified?

2) The images look as though transfection efficiency is not 100%, which likely dampens the impact on expression differences between negative control and dCas9-DNMT1 plasmids, which may be especially relevant if the levels of PRAME-AS are already low. Can the GFP+ cells be enriched via cell sorting to parse this out, or can the authors speak to this possibility?

3) The title of the figure states that hypermethylation diminishes PRAME-AS transcripts yet the quantification in Figure 5D is not significant, this should be corrected or clarified.

Figure 6

Here the authors are attempting to determine whether PRAME-AS is the result of a bidirectional promoter (from PRAME). Here, the luciferase results from Schenk et al (2007) are used as a marker for determining where the maximum promoter activity for PRAME is located. In that paper, PRAME is shown to have two transcription start sites (TSS) annotated, in which the downstream one has stronger activity, and may be the source of the bidirectionally of PRAME-AS, thus the overlap of the 5’ exon is due to annotation of the upstream PRAME TSS with the antisense transcription of PRAME-AS resulting from the downstream PRAME TSS. Can the authors speak to this possibility?

Bidirectional transcription from protein-coding genes often contain promoter proximal polyadenylation signals in the antisense direction (in this case PRAME-AS). Can the authors confirm whether the “PRAME-AS direction” construct contains any AATAAA/ATTAAA signals that may prematurely terminate transcription upstream of GFP?

A much more convincing demonstration of bidirectional promoter activity would be to only include the CpG island region as shown in Figure 5 (perhaps overlapping PRAME promoter as well), if possible even including +/- in vitro construct methylation by SssI as shown in Shenk et al. Alternatively, can it be explained in more detail as to how this region is not "bidirectional" based on the tests that have already been completed?

Minor Comments

1) Figure 1: Font of y-axis is very small, should be enlarged to scale with the rest of the graph

2) Figure 1: It is very interesting that PRAME-AS appears to be completely silenced in testicular germ cell tumors. What is the p-value of the comparison between testicular germ-cell tumors and normal testicular cells

3) It is fine to reference a prior paper for details on the methods regarding TransCRISTI, however the methods section of this publication should still have a section dedicated to it, even if it is only to refer to the prior paper.

4) Figure 5: Can it be labeled more explicitly which sgRNA corresponds to which CpG island?

2) Figure 4: What happens to PRAME and PRAME-AS expression if MZF1 is upregulated in PRAME-AS KO cells? Is there a link between MZF1 recruitment and PRAME-AS influence on PRAME?

5) Figure 5: Could the authors show that reducing DNA methylation at the PRAME-AS ‘promoter’ leads to increased PRAME-AS expression i.e through dCas9-Tet1?

**Reviewer #3: ** Overview

In this study, the authors investigate the effects of long non-coding RNA (LncRNA) PRAME-AS gene on the expression of PRAME, cell proliferation, migration, cell viability and anchorage-independent growth with results showing a decrease in all of these functions. Upregulation of PRAME-AS via MZF1 overexpression and hypermethylation lead to an increase in the above-mentioned characteristics.

Overall, this study was written clearly and the work was well performed.

Comments

Major comments:

1. A full description of the TCGA and GTEx datasets for bioinformatics analyses should be included in M&M.

2. Some of the results sections describe details of the technique used more than an elaborate description of the actual results and their significance.

3. More focus on the significance of the findings is needed in the discussion.

Minor comments:

Tables 1 & 2 can be added to supplementary information.

**Reviewer #4: ** Hosseininia and Dehghani investigated the role of the antisense lncRNA of the PRAME gene (PRAME-AS) in PRAME regulation and cell phenotypes. The authors revealed that i) PRAME-AS knockout decreased PRAME mRNA levels by approximately 34%, ii) cell migration, viability, and stemness were decreased in PRAME-AS knockout cells compared to control cells, iii) MZF1 increased the expression of PRAME-AS and PRAME transcripts, and iv) DNA hypermethylation of PRAME-AS upstream downregulated the expression of PRAME-AS and PRAME transcripts.

The authors concluded that PRAME-AS regulates PRAME at the transcriptional level. However, the data provided are insufficient to support this conclusion.

Major Comments:

1. Although the authors used HEK293 cells to investigate the effects of PRAME-AS on cell stemness, these cells were not optimal for the stemness assay due to their insufficient colony size for demonstrating differences. Furthermore, the function of PRAME is well-established in cancer cells. The authors should use at least one cancer cell line in addition to HEK293 cells. The utilization of more than two cell lines would be preferable for generalizing their findings.

2. The authors should investigate the effect of PRAME-AS on cell growth. The number of control and PRAME-AS knockout cells should be counted from the initial day of culture until day 3 or 5.

3. Despite being statistically significant, the observed reduction in PRAME mRNA levels and the diminished stemness and migratory capabilities resulting from PRAME-AS knockout appear to be minimal. These subtle changes may not result in substantial biological effects. The authors should provide an explanation for the modest decrease in PRAME mRNA and elucidate why there are only slight differences between the wild-type cells and those with PRAME-AS knockout in both the soft agar and scratch assays (Figures 2G, 3C, and 3D).

4. Although the authors avoided CpG island on the primers for bisulfite sequencing (page 10, line 162), DNA methylation of CpG island is crucial for the regulation of gene expression. Primers should be designed to analyze CpG methylation on CpG islands.

5. Figure 6A should be carefully edited for the following reasons:

a) Although the authors stated that H3K27ac data of the PRAME promoter was from UCSC Genome Browser (page 25, line 495), the observed H3K27ac peak differed from the current information available on the UCSC Genome Browser (https://genome.ucsc.edu/cgi-bin/hgTracks?db=hg38&lastVirtModeType=default&lastVirtModeExtraState=&virtModeType=default&virtMode=0&nonVirtPosition=&position=chr22%3A22556888%2D22562353&hgsid=2476779099_S0v0GSEFx5vKwQS2lhUHcGN4aBHD).

b) The positions of CpG island are based on a previous study (Schenk T et al, ref#22). However, this is the predicted position, and the current official annotation of CpG island locations can be found in the UCSC Genome Browser.

c) It is unclear how to generate the continuous peak of PRAME promoter activity from the intermittent data obtained through the luciferase assay.

6. The authors should carefully edit the entire manuscript:

a) The Discussion section needs revision to eliminate redundancies and excessive length; for examples, the detailed descriptions of other antisense lncRNA are not necessary (page 27, lines 534 – 548), the detailed information of MZF1 (page 29, lines 574 – 595) should be shorten because the regulation of PRAME by MZF1 has been reported in previous papers (Lee et al, ref#23 etc.), repeated description of the results are not necessary (page 30, lines 612 – 616).

b) “CPG” should be changed to “CpG” throughout the manuscript.

6. PLOS authors have the option to publish the peer review history of their article (what does this mean? ). If published, this will include your full peer review and any attached files.

**Do you want your identity to be public for this peer review?** For information about this choice, including consent withdrawal, please see our Privacy Policy .

Reviewer #1: No

Reviewer #2: **Yes: ** Michael Robert Murphy

Reviewer #3: No

Reviewer #4: No

---

## [Author Response · Author response to Decision Letter 1]

30 Jun 2025

We would like to express our sincere gratitude to the reviewers for their time and thoughtful comments on our manuscript. Please find a point-by-point reply to the reviewers’ comments below, and note that we have included the unedited comments of the reviewers in italic.

Reviewer 1

This study demonstrates that a lncRNA (PRAME-AS) can regulate its homologous coding gene PRAME. The authors identify MZF1 and hypermethylation of the upstream regulatory region as regulators of this lncRNA's transcription, and ultimately reveal that the PRAME promoter functions unidirectionally in regulating PRAME. While this discovery is novel, the main findings require further experimental validation to strengthen their impact.

Main concerns:

1. The title emphasizes the lncRNA and expression of PRAME, but the narrative lacks cohesion: The authors first identify the lncRNA, after knock out, the PRME expression is reduced, how does LncRNA regulate PRME? without explore further, the author turned to check cell proliferation, migration, viability, and anchorage-independent growth, then they turn to the explore its upstream regulatory factors (MZF1 and hypermethylation),and later shift to analyzing bidirectional promoter activity, Please ensure that all sections are connected naturally and smoothly.

Response: We thank the reviewer for this point. The main message of the manuscript is that PRAME-AS lncRNA can function as a regulator of PRAME transcript levels. We checked every instance regarding the relationship between these two genes throughout the manuscript and carefully corrected them to convey this message. Different sections of the manuscript that include these changes are listed below.

Title:

PRAME-AS lncRNA, Regulated by MZF1, Modulates PRAME Expression and Cell Stemness

Abstract:

Taken together, our data support the conclusion that PRAME-AS lncRNA acts as a regulator of PRAME transcript levels, and that MZF1—and the methylation status of a shared CpG island—influences the expression of both genes.

Introduction:

Overall, our data suggest that PRAME-AS lncRNA can be regarded as a regulator of PRAME transcript levels.

Results:

Subheading #2 and Figure 2 legend: Ablation of the PRAME-AS gene affects PRAME transcript levels

Discussion:

In this study, utilizing a knockout strategy, we demonstrate that the PRAME-AS lncRNA functions as a regulator of PRAME transcript levels (Fig 2).

This is the first study that investigates the PRAME-AS and the effects of its ablation on PRAME transcript levels (Fig 2).

We have edited the “Results” subheadings and added/changed explanations to provide a more cohesive narrative for this study.

2. The abstract uses non-definitive phrases ("lncRNA MIGHT regulate PRAME", "MZF1 MAY affect both genes"). If the results are reliable, replace tentative terms ("might/may") with stronger language (e.g.,"demonstrates" or "mediates") to enhance scientific rigor.

Response: We have corrected the abstract according to the reviewer’s point.

3. PRAME-AS lncRNA transcripts show no significant differential expression between normal and tumor tissues. This raises concerns about its functional role in tumor progression. what drives authors continues to explore this LncRNA in this situation?

Response: A lack of significant differential expression between normal and tumor tissues does not automatically undermine the functional relevance of an lncRNA. LncRNAs may exhibit crucial roles while not exhibiting differential expression. In addition, despite a subtle differential expression, changes in post-translational modifications, RNA stability, splicing patterns, cellular localization, or interactions with proteins could also influence their cellular functions.

4. The study attributes PRAME-AS-mediated phenotypes (proliferation, migration, etc.) to lncRNA activity, but it remains unclear whether these effects are PRAME-dependent or not. To confirm that the observed phenotypes are specifically regulated by PRAME-AS (and not solely by PRAME), include PRAME knockdown/knockout controls in functional assays.

Response: We thank the reviewer for this important point. It is very difficult—if not impossible—to directly relate the effects of a gene knockout to specific cell phenotypes (i.e., cell functions). The primary objective of our study was to knock out the antisense lncRNA gene and assess its impact on cell proliferation, viability, and stemness. We did not intend to attribute these effects to PRAME or PRAME-AS, nor have we stated anywhere in the manuscript that these phenotypes result from PRAME dysregulation.To clear this objective, we have changed and corrected the text throughout the manuscript accordingly:

Abstract:

The PRAME-AS knockout cells showed a decrease in migration, proliferation, stemness, and viability.

Results:

Subheading # 3 and Figure 3 legend: The PRAME-AS knockout cells showed a decrease in migration, proliferation, stemness, and viability

Minor points

1. Figure 1: Include PRAME protein expression to complement transcript-level findings. This could be added as supplementary data.

Response: A supplementary figure 1 (S1 Fig) containing the protein expression levels in different tumors has been added.

2. Figure 2G: Provide PRAME protein expression results following PRAME-AS knockout to validate transcriptional regulation at the protein level.

Response: We thank the reviewer for this point. We did not have access to the PRAME antibody. Thus, we kept our focus on the effects of PRAME-AS lncRNA on PRAME transcript levels.

3. Figure 4 Include Western blot data to confirm successful MZF1 overexpression.

Response: Since one of the defining features of MZF1 in a cell is its nuclear localization, and we have used these overexpressing cells in a separate study, we chose to perform immunocytochemical staining instead of Western blotting to confirm MZF1 overexpression and its nuclear localization after transposon-mediated gene transfer. Our results (Fig 4 and S8 Fig.) showed that cells after transposition, compared to control cells, had strong nuclear signals.

4. Figure 6C Clearly the cell number of AS direction is much less than the other two.

Response: We thank the reviewer for this point. In Fig 6C, we replaced the panel of cells and used a better representative image.

Reviewer 2

In this paper, titled “PRAME-AS lncRNA is regulated by MZF1 and is involved in the expression of PRAME transcripts and cell stemness”, the authors have undertaken a characterization of a previously unstudied lncRNA originating antisense from tumor antigen PRAME (PRAME-AS). Regulation of PRAME-AS has been analyzed via interrogation of known expression datasets, transcription factor overexpression, the efficacy of its promoter region, and the extent of CpG methylation surrounding it. The function of PRAME-AS was assessed via Cas9-mediated ablation with downstream impacts on PRAME expression, proliferation, and cell migration.

It is remarkable that for a gene with long known implications in tumorigenesis that the antisense lncRNA expressed at this locus has so far gone uncharacterized and so the authors should be commended for tackling this as a research topic. The method of PRAME-AS KO so as not to alter promoter activity of PRAME is elegant, however there are several concerns regarding the veracity of the conclusions until further clarity is reached.

Response: We sincerely appreciate the reviewer’s insightful summary and positive evaluation.

Major comments:

Figure 2

1) What is the expression level of PRAME-AS in HEK293T cells, and how does it compare to PRAME expression approximately? Low expression levels are not necessarily an impediment to study as lncRNAs are commonly lowly expressed, but it should be made clear relative to the housekeeping gene.

Response: We have included the expression levels for PRAME-AS in wild-type cells. However, as we discussed and according to our previous and recent experiments, which have been incorporated in the “Results (Fig. 2F, Fig. 2G, and Fig. 2H)”, the PRAME-AS transcript was not detectable in PRAME-AS knockout cells. We have added a new panel G to show that the PRAME-AS is not detectable by RT-PCR. We have also incorporated the expression level of PRAME-AS in wild-type cells to Panel H.

2) What is the extent of PRAME-AS decrease after integration of the donor fragment? Preferably shown with primers both upstream (Exon 2) and downstream (Exon 3) of the integrant.

Response: As explained in the previous answer, we were not able to detect PRAME-AS RNA after integration of the donor fragment Fig. 2G) when we used primers on either side of the integration point.

3) There is concern that inserting a construct inversely to PRAME-AS but on the same strand and upstream as PRAME may lead to interference with PRAME transcription if there is transcription readthrough of the construct.

Response: We thank the reviewer for this important point. In the original Figs 2B and 2C, we had mistakenly omitted the polyA signal. Now, the revised Fig. 2B and C show these signals, indicating that the transcription started from the CAG promoter will certainly stop at this point. In addition, between the end of the integrated transgene and the transcriptional start site of the PRAME, there is a 6,063 bp distance, which makes it very unlikely that the PRAME promoter would be affected by the transgene.

a) Does the sequence have (a) polyadenylation signal(s) to ensure transcription termination of the integrant? b) Can it be shown that there is no transcription readthrough downstream of the 3’TR region?

Response: The response to these questions has been provided in the above description.

4) Figure 2F: The presence of the Puro-IRES-GFP integrant is demonstrated, however it isn’t clear or explicitly stated whether this is integrated into or one both alleles. Can this be shown or clarified?

Response: We thank the reviewer for this question. After integration of the CAG-Puro-IRES-EGFP-polA integrant, cells were subjected to several rounds of puromycin treatment. However, we did not grow cells clonally to be able to check them for mono-allelic or bi-allelic integration. We have incorporated a new “TransCRISTI method” section to Materials and Methods. The section is read as:

TransCRISTI method

To ablate the PRAME-AS lncRNA gene, we used our previously developed TransCRISTI method [28]. Briefly, HEK293T cells were transfected (using 25 kDa branched Polyethylenimine; bPEI25, Sigma-Aldrich, USA) with Cas9.PBdm (Cas9 fused to double mutant piggyBac transposase) and sgRNA #1 encoding plasmid (pHD_5009-1), and a donor plasmid (pHD_4012; 5’ TR-CAG promoter-PuroR-IRES-EGFP-PA-3’TR) (Table S1). After 2 hours of incubation with polyplex, the transfection medium was replaced with the medium containing 10% FBS and no antibiotics. Forty-eight hours post-transfection, cells were selected for 3 days in the medium containing 0.7 μg/ml puromycin sulfate (Sigma-Aldrich, USA), followed by a 3-day recovery (medium without puromycin sulfate). Periods of puromycin treatment and recovery were continued until we had an enriched EGFP-expressing population of cells.

28. Rezazade Bazaz M, Ghahramani Seno MM, Dehghani H. Transposase-CRISPR mediated targeted integration (TransCRISTI) in the human genome. Sci Rep. 2022; 12(1):3390. PMID: 35232993.

Figure 3

1) It would be ideal to perform a rescue experiment of overexpressing PRAME in the PRAME-AS KO cells to determine whether those phenotypes are due to the ~34% loss of PRAME.

Response: We did not have access to a PRAME-expressing construct to perform this experiment. However, in several studies, as discussed in “Discussion”, the PRAME overexpression has been linked to increased cell proliferation, cell stemness, and cell viability. In addition, as we explained in the answer to question 4 of Reviewer 1, we do not think that these cell phenotypes can all be related to the effects of a single gene (here, PRAME), and thus, overexpression (rescue) experiments and their effects on cell phenotypes would not be ideal to study the effects of this knouckout further. Our primary objective was to demonstrate whether or not these cell functions are affected in response to the PRAME-AS knockout. We did not intend to label these effects as PRAME- or PRAME-AS-dependent. Related paragraph in “Discussion”:

As previously indicated, there is strong evidence supporting the role of PRAME as a key contributor to cellular malignancy across various cancer types. For example, in triple-negative breast cancer (TNBC), PRAME overexpression induces the expression of 11 EMT (epithelial-to-mesenchymal transition)-related genes and leads to deregulation of Notch and Wnt signaling pathways, inhibition of apoptosis, and degradation of basement membranes in blood vessels [8]. Moreover, PRAME is elevated in hepatocellular carcinoma and prohibits apoptosis mediated by p53 and Bcl2 [9]. High level of PRAME has a strong positive correlation with the EMT-related transcription factors SNAI1 and TWIST, while normal liver regulators, including FOXA2 and AR, are negatively correlated with PRAME expression [26]. PRAME promotes the proliferation of osteosarcoma cells [10], while in Chronic myelogenous leukemia, in the presence of RA and PcG proteins, including EZH2, it binds to retinoic acid receptor (RAR) to block Caspase 3, TRAIL, and P21 expression, leading to proliferation and apoptosis arrest [14,15]. Another study in seminomas indicates that the overexpression of PRAME and its interaction with Lin28A support pluripotency networks and expression of reprogramming markers, such as DPPA3, Nanog, Oct3/4, ZSCAN10, and PRDM14 [20]. Two distinct research reports on non-small cell lung cancer (NSCLC) patients of Asia (South Korea, Singapore, Taiwan, and Thailand) showed a higher rate of PRAME transcripts in squamous cell carcinoma (SCC), and in smokers compared with non-smokers [16,17]. A recent study on cervical cancer indicates PRAME’s involvement in proliferation, invasion, migration, reduction of apoptotic cells, and facilitating G0/G1 cell-cycle by the Wnt/β-Catenin pathway [21]. Another study supports the role of PRAME in proliferation, invasion, migratory potential, and EMT in LSCC, and the growth of tumors in response to PI3K/AKT/mTOR pathway activation [27]. These reports highlight the involvement of diverse genetic and epigenetic factors, regulatory networks, and pathways downstream of PRAME that influence cellular functions such as proliferation, viability, and stemness. In this study, we expand the players to add the PRAME-AS lncRNA to this regulatory network. However, since in this study, we manipulated only the PRAME-AS gene, we cannot expect to observe considerable changes in all kinds of cellular functions (Fig 3). For example, in the PRAME-AS knockout cells, MZF1, which is a positive regulator of PRAME, is still expressed and will continue to play its regulatory role.

Figure 4

1) Related to Figure 2, the fold change of PRAME and PRAME-AS can be seen but how do they compare to each other? It would be very interesting to see similar fold increases when overexpressing MZF1 if there were a large basal difference between PRAME and PRAME-AS.

Response: To show the basal difference between PRAME and PRAME-AS transcript levels, we included a supplementary S9 Figure, which has been prepared based on the copy number.

Figure 5

1) If PRAME-AS expression levels are already low, and methylation levels high, then any dCas9 methylation will likely not alter PRAME-AS expression levels further. Can the level of methylation of each of the CpGs analyzed with bisulfite sequencing be quantified?

Response: To answer this question, we analyzed the methylation status of CpG dinucleotides in the HEK293T wild type (before transfection) and added this result to Fig. 5D.. As the sequencing results show, the CpGs were methylated; however, after utilizing the dCas9-DNMT3A, the amount of methylation was increased.

2) The images look as though transfection efficiency is not 100%, which likely dampens the impact on expression differences between negative control and dCas9-DNMT1 plasmids, which may be especially relevant if the levels of PRAME-AS are already low. Can the GFP+ cells be enriched via cel

---

## [Decision Letter · Decision Letter 1]

20 Jul 2025

PONE-D-25-03086R1PRAME-AS lncRNA, regulated by MZF1, modulates PRAME expression and cell stemnessPLOS ONE

Dear Dr. Dehghani,

Thank you for submitting your manuscript to PLOS ONE. After careful consideration, we feel that it has merit but does not fully meet PLOS ONE’s publication criteria as it currently stands. Therefore, we invite you to submit a revised version of the manuscript that addresses the points raised during the review process.

Your manuscript was reviewed by three referees who had reviewed your original one. Basically, they are satisfied with your revision, but one of them suggested some minor issues to be resolved. Please revise it according to their suggestions.

We look forward to receiving your revised manuscript.

Kind regards,

Hodaka Fujii, M.D., Ph.D.

Academic Editor

PLOS ONE

Journal Requirements:

Reviewers' comments:

Reviewer's Responses to Questions

**Comments to the Author**

1. If the authors have adequately addressed your comments raised in a previous round of review and you feel that this manuscript is now acceptable for publication, you may indicate that here to bypass the “Comments to the Author” section, enter your conflict of interest statement in the “Confidential to Editor” section, and submit your "Accept" recommendation.

Reviewer #2: All comments have been addressed

Reviewer #3: All comments have been addressed

Reviewer #4: All comments have been addressed

2. Is the manuscript technically sound, and do the data support the conclusions?

Reviewer #2: Yes

Reviewer #3: Yes

Reviewer #4: Yes

3. Has the statistical analysis been performed appropriately and rigorously? 

Reviewer #2: Yes

Reviewer #3: Yes

Reviewer #4: Yes

4. Have the authors made all data underlying the findings in their manuscript fully available?

Reviewer #2: Yes

Reviewer #3: Yes

Reviewer #4: Yes

5. Is the manuscript presented in an intelligible fashion and written in standard English?

Reviewer #2: Yes

Reviewer #3: Yes

Reviewer #4: Yes

6. Review Comments to the Author

Reviewer #2: Minor comments for Hosseininia & Dehghani:

1. Figure 2A spelling: “PRAM-AS”

2. Wording regarding unidirectionality – PRAME-AS transcription appears to be dependent on PRAME transcription. I don’t think it is correct to say that “PRAME promoter operates unidirectionally, regulating only PRAME transcription”. PRAME promoter clearly regulates PRAME-AS as well through bidirectional transcription. Perhaps more accurate to say that PRAME-AS strand promoter region is insufficient for transcription (based on reporter expression) without PRAME transcription (the PRAME strand) to drive its expression.

3. In the discussion it is mentioned that a possible regulatory mechanism may involve direct interactions between PRAME and PRAME-AS via RNA-RNA duplex. A BLAST search of spliced PRAME-AS compared to PRAME entire transcript (exon and intron) seems to indicate that ~100nt of PRAME-AS exon 3 has complementarity with PRAME intron. If RNA-RNA duplex is mentioned as a possibility, then evidence or lack of evidence of this through sequence search should be done.

Reviewer #3: (No Response)

Reviewer #4: (No Response)

7. PLOS authors have the option to publish the peer review history of their article (what does this mean? ). If published, this will include your full peer review and any attached files.

**Do you want your identity to be public for this peer review?** For information about this choice, including consent withdrawal, please see our Privacy Policy .

Reviewer #2: **Yes: ** Michael R. Murphy

Reviewer #3: No

Reviewer #4: No

---

## [Author Response · Author response to Decision Letter 2]

6 Aug 2025

Reviewer #2: Minor comments for Hosseininia & Dehghani:

1. Figure 2A spelling: “PRAM-AS”

Response: We thank the reviewer for pointing this out. We have corrected this typo, along with a few others that were found in other figures.

2. Wording regarding unidirectionality – PRAME-AS transcription appears to be dependent on PRAME transcription. I don’t think it is correct to say that “PRAME promoter operates unidirectionally, regulating only PRAME transcription”. PRAME promoter clearly regulates PRAME-AS as well through bidirectional transcription. Perhaps more accurate to say that PRAME-AS strand promoter region is insufficient for transcription (based on reporter expression) without PRAME transcription (the PRAME strand) to drive its expression.

Response: We agree that the potential involvement of the PRAME promoter in the transcriptional regulation of PRAME should be taken into account. This possibility has already been discussed in the “Discussion” section:

However, our reporter assay that includes the PRAME locus regulatory region did not reveal a considerable promoter function in the PRAME-AS direction (Fig 6). In support of our findings, a recent study reported that a ~300-nucleotide common regulatory region with signatures of a bidirectional promoter for the genes LINC00882 and DUBR, lost its promoter activity when upstream regions were also cloned next to this common region [66]. Thus, one possible explanation for our findings is that the promoter activity of the defined region may have been negatively affected by additional upstream sequences.

Per the reviewer’s suggestion, we have revised the sentence in the “Abstract”, “Results” and Figure 6 legend as follows:

Abstract and Results

Based on our EGFP reporter assay, we conclude that the regulatory region of the PRAME locus—without PRAME transcription—is insufficient to drive transcription of PRAME-AS in the antisense direction.

Figure 6 legend

The PRAME locus regulatory region regulates transcription of a reporter gene only in the PRAME direction

3. In the discussion it is mentioned that a possible regulatory mechanism may involve direct interactions between PRAME and PRAME-AS via RNA-RNA duplex. A BLAST search of spliced PRAME-AS compared to PRAME entire transcript (exon and intron) seems to indicate that ~100nt of PRAME-AS exon 3 has complementarity with PRAME intron. If RNA-RNA duplex is mentioned as a possibility, then evidence or lack of evidence of this through sequence search should be done.

Response: We appreciate the reviewer's important point. To examine potential interactions between spliced PRAME-AS and PRAME mRNA, we used several tools, including LncTar, LncRRIsearch, ENCORI, RNAInter, and NPInter v5.0. However, none of these tools predicted any interactions. We also performed BLAST analysis and found that exon 3 of PRAME-AS shares sequence similarity with a 256-base region within intron 3 of the PRAME gene. Additionally, exon 1 of PRAME-AS shows similarity to a 65-base region in the 5′ UTR of the PRAME gene, located 77 bases upstream of the transcription start site (TSS). However, we note that BLAST analysis identifies sequence similarity rather than complementarity.

Thus, considering the above explanations and the multitude of mechanisms through which lncRNAs can influence mRNA levels, we believe that concluding the “Discussion” section with a sentence indicating that further experiments are needed to elucidate the underlying mechanisms would be appropriate.

---

## [Decision Letter · Decision Letter 2]

13 Aug 2025

PRAME-AS lncRNA, regulated by MZF1, modulates PRAME expression and cell stemness

PONE-D-25-03086R2

Dear Dr. Dehghani,

We’re pleased to inform you that your manuscript has been judged scientifically suitable for publication and will be formally accepted for publication once it meets all outstanding technical requirements.

Kind regards,

Hodaka Fujii, M.D., Ph.D.

Academic Editor

PLOS ONE

Additional Editor Comments (optional):

Reviewers' comments:

Reviewer's Responses to Questions

**Comments to the Author**

1. If the authors have adequately addressed your comments raised in a previous round of review and you feel that this manuscript is now acceptable for publication, you may indicate that here to bypass the “Comments to the Author” section, enter your conflict of interest statement in the “Confidential to Editor” section, and submit your "Accept" recommendation.

Reviewer #2: All comments have been addressed

2. Is the manuscript technically sound, and do the data support the conclusions?

Reviewer #2: Yes

3. Has the statistical analysis been performed appropriately and rigorously? 

Reviewer #2: Yes

4. Have the authors made all data underlying the findings in their manuscript fully available?

Reviewer #2: Yes

5. Is the manuscript presented in an intelligible fashion and written in standard English?

Reviewer #2: Yes

6. Review Comments to the Author

Reviewer #2: (No Response)

7. PLOS authors have the option to publish the peer review history of their article (what does this mean? ). If published, this will include your full peer review and any attached files.

**Do you want your identity to be public for this peer review?** For information about this choice, including consent withdrawal, please see our Privacy Policy .

Reviewer #2: **Yes: ** Michael R. Murphy

---

## [Editor Report · Acceptance letter]

PONE-D-25-03086R2

PLOS ONE

Dear Dr. Dehghani,

I'm pleased to inform you that your manuscript has been deemed suitable for publication in PLOS ONE. Congratulations! Your manuscript is now being handed over to our production team.

Kind regards,

on behalf of

Dr. Hodaka Fujii

Academic Editor

PLOS ONE